# Visualizing group II intron dynamics between the first and second steps of splicing

Jacopo Manigrasso [1,8], Isabel Chillón [2,8], Vito Genna[3], Pietro Vidossich[1], Srinivas Somarowthu[4], Anna Marie Pyle[5,6,7], Marco De Vivo [1✉] & Marco Marcia [2✉]

Group II introns are ubiquitous self-splicing ribozymes and retrotransposable elements evolutionarily and chemically related to the eukaryotic spliceosome, with potential applications as gene-editing tools. Recent biochemical and structural data have captured the intron in multiple conformations at different stages of catalysis. Here, we employ enzymatic assays, X-ray crystallography, and molecular simulations to resolve the spatiotemporal location and function of conformational changes occurring between the first and the second step of splicing. We show that the first residue of the highly-conserved catalytic triad is protonated upon 5'-splice-site scission, promoting a reversible structural rearrangement of the active site (toggling). Protonation and active site dynamics induced by the first step of splicing facilitate the progression to the second step. Our insights into the mechanism of group II intron splicing parallels functional data on the spliceosome, thus reinforcing the notion that these evolutionarily-related molecular machines share the same enzymatic strategy.

---

[1] Laboratory of Molecular Modelling & Drug Discovery, Istituto Italiano di Tecnologia, Via Morego 30, 16163 Genoa, Italy. [2] European Molecular Biology Laboratory (EMBL) Grenoble, 71 Avenue des Martyrs, Grenoble 38042, France. [3] Department of Structural and Computational Biology, Institute for Research in Biomedicine (IRB), Parc Científic de Barcelona, C/ Baldiri Reixac 10-12, 08028 Barcelona, Spain. [4] Department of Biochemistry & Molecular Biology, Drexel University College of Medicine, Philadelphia, PA, USA. [5] Department of Molecular, Cellular and Developmental Biology, New Haven, CT 06511, USA. [6] Department of Chemistry, Yale University, New Haven, CT 06511, USA. [7] Howard Hughes Medical Institute, Chevy Chase, MD 20815, USA. [8] These authors contributed equally: Jacopo Manigrasso, Isabel Chillón. ✉email: marco.devivo@iit.it; mmarcia@embl.fr

Self-splicing group II intron ribozymes are essential regulators of gene expression in all domains of life and they share evolutionary origins and enzymatic properties with the spliceosome, the eukaryotic machinery that catalyzes nuclear splicing of mRNA precursors[1,2]. Spliced group II introns are active retrotransposable elements that contribute to genomic diversification with potential applications in medicine and gene editing[3,4]. Therefore, elucidating the mechanism of group II intron catalysis is crucial for understanding key steps in gene expression and RNA maturation, and to develop therapeutic and biotechnological tools.

The current understanding of group II intron self-splicing mechanism derives from biochemical and cell biology studies[5–8] and from 3D structures of introns from various phylogenetic classes[9–14]. These studies have provided detailed molecular insights on intron folding[15] and high-resolution molecular snapshots of the *Oceanobacillus iheyensis* group IIC intron trapped in various conformations throughout the catalytic cycle[16–20].

The intron catalytic site comprises a highly conserved triple helix formed by nucleotides of the so-called catalytic triad (in domain D5, C358-G359-C360), two-nucleotide bulge (D5, A376-C377), and J2/3 junction (between D2 and D3, A287-G288-C289, all numbering from the crystallized form of the *O. iheyensis* intron, i.e. PDB id: 4FAQ; Supplementary Fig. 1a, b). The site also harbors a metal-ion cluster formed by two divalent (M1–M2) and two monovalent (K1–K2) ions (Supplementary Fig. 1a). These ions participate directly in catalysis[16,19], which occurs via a series of nucleophilic $S_N2$ reactions (Fig. 1a). In the first step of splicing, depending on whether the intron follows a hydrolytic or a transesterification mechanism, respectively[21], a water molecule or the 2′-OH group of a bulged adenosine in D6, activated by M2 and by the triple helix, attack the 5′-splice junction of the precursor (5e-I-3e), forming an intron/3′-exon intermediate (I-3e), in which the scissile phosphate (SP) is coordinated by K2. In the second step of splicing, the 5′-exon (5e), activated by M1, performs a nucleophilic attack on the 3′-splice junction, releasing ligated exons (5e-3e) and a linear or lariat form of the excised intron (I; Fig. 1a). The latter can then further reverse splice into cognate or non-cognate genomic DNA, in processes known as retrohoming or retrotransposition[22,23]. Crystal structures of the pre- and post-hydrolytic states are available for the first and second steps of splicing, allowing precise localization of reactants[13,16], and computational studies have elucidated energetics and dynamics of the related reaction chemistry[24,25].

However, a key aspect of the group II intron splicing cycle that remains largely uncharacterized is the transition between the splicing steps, when the intron must release products of the first reaction and recruit substrates of the second splicing event. Biochemical and structural studies suggest that, after the first step of splicing, the intron rearranges at the K1-binding site, transiently adopting a specific inactive conformation (aka the toggled conformation), in which G288 (in the J2/3 junction) and C377 (in the two-nucleotide bulge) disengage from their triple helix with the catalytic triad of nucleotides in D5, thereby disrupting the catalytic metal center[16,26] (Supplementary Fig. 1a). Parallel studies also suggest that group II intron conformational changes may be triggered by protonation of active site nucleotides during the splicing cycle[27]. Specifically, the N1 atom of adenosines (N1A) and the N3 atom of cytosines (N3C) can undergo large $pK_A$ shifts in folded DNA or RNA and thereby serve as proton donors/acceptors, much like histidine residues in proteins[28]. Consistent with this, functional studies on the spliceosome suggest that protonation within the U6 intramolecular stem-loop (ISL), which is analogous to the group II intron two-nucleotide bulge and catalytic triad, antagonizes binding of catalytic metal ions and induces transient base-flipping during splicing[29,30].

To understand the transition between the first and second step of splicing, here we probe the group II intron active site by mutagenesis, enzymatic assays, crystallography, and molecular dynamics (MD) modeling. We find that, immediately after the first step of splicing, protonation of a conserved nucleobase within the catalytic triad promotes the spontaneous release of K1 and induces intron toggling. Consistent with this finding, intron mutants that cannot be protonated have defects in the second step of splicing. Our group II intron data have parallels with functional studies on the nuclear spliceosome, suggesting that protonation and toggling are common mechanistic strategies that are adopted by both these splicing machines.

## Results

**A catalytic residue may become protonated during splicing.** Because crystal structures of distinct states of the *O. iheyensis* group II intron are available, we first analyzed these structures using continuum electrostatics to obtain an initial qualitative approximation of the $pK_A$ values of active site nucleotides (Supplementary Table 1). Using nonlinear Poisson–Boltzmann calculations, we noted that the $pK_A$ value of most residues remains unchanged (Supplementary Table 1). By contrast, the computed $pK_A$ value of C358 (catalytic triad) shifts between the pre-hydrolytic state ($pK_A \sim 4.5$ in PDB id: 4FAQ) and the so-called toggled state that forms after the first step of splicing[16] ($pK_A \sim 7.2$ in PDB id: 4FAU). Although these values are qualitative due to the influence of geometrical changes and uncertainties in the definition of the grid and dielectric constants, the Poisson–Boltzmann calculations suggest that C358 has different protonation states along the splicing trajectory (Supplementary Fig. 1c). Consistent with these findings, nucleotide position 358 in other introns can be occupied by an adenine or a cytidine, i.e., bases that can be protonated, but this same position never varies to guanidine or uracil, i.e., bases that cannot be protonated[31].

Computational studies on the *O. iheyensis* group II intron immediately after 5e hydrolysis have identified proton transfer pathways from the reaction nucleophile into the bulk solvent involving up to five water molecules (corresponding to a migration distance of ~15 Å)[24]. Although less efficient than direct proton transfer, such chains of water molecules enable a proton to shuttle from the nucleophile to the N3 atom of C358 ($N3^{C358}$), which is exposed within the same solvent-filled cavity at a distance of 9.8 Å in the structure of the pre-hydrolytic state (PDB id: 4FAQ)[32] (Supplementary Fig. 2a). Moreover, our hybrid quantum (DFT/BLYP)/classical simulations show that, once a proton is positioned at the N3 atom, C358 remains stably protonated for over 15 ps (see "Methods" section and Supplementary Fig. 2b, c).

Taken together, our observations from continuum electrostatics and quantum mechanical (QM) simulations, the specific evolutionary conservation pattern of C358, and its key structural role in the pre-hydrolytic state suggested that C358 plays a direct role in group II intron catalysis.

**Non-protonatable mutants show second step splicing defects.** To explore the functional role of C358 in reaction chemistry, we created *O. iheyensis* splicing precursor constructs[16] in which C358 was replaced with A, G, or U. In addition, to maintain the structural integrity of the catalytic triple helix, we isosterically replaced the two partners of C358, i.e., its Watson–Crick pairing partner (position 385) and its J2/3 triple-helical partner (position 289) (Supplementary Fig. 1d). After incorporating the resulting triple base mutations (C289A/C358A/G385U, aka the A-mutant; C289G/C358G/G385C, aka the G-mutant; and

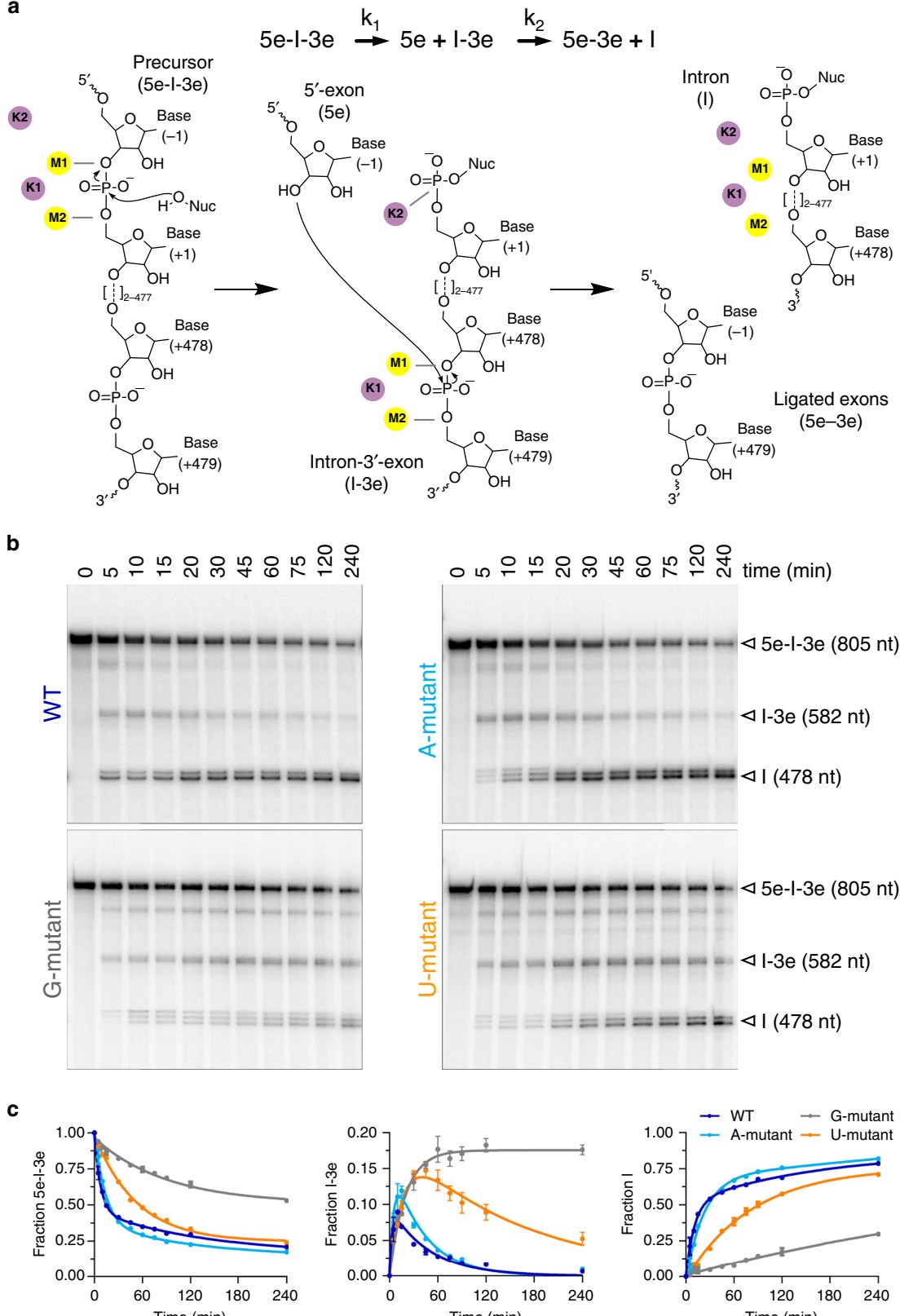

**Fig. 1 Kinetics of intron mutants. a** Schematics of the splicing reaction and sketch of the chemical mechanism of catalysis by group II introns. $k_1$ is the rate constant of the first and $k_2$ of the second step of splicing. Kinetic rate constants of all constructs are reported in Supplementary Table 2. Black arrows indicate nucleophilic attacks; gray dotted lines indicate interactions between oxygen atoms of the scissile phosphate groups and catalytic metal ions; Nuc indicates the reaction nucleophile. **b** Representative splicing kinetics of wild-type intron and A, G, and U mutants. Precursors are indicated as 5e-I-3e (nt length in parenthesis). Intermediate (I-3e) and linear intron (I) migrate as double or triple bands because of cryptic cleavage sites, as explained previously[16]. **c** Evolution of the populations of precursor (5e-I-3e, left panel), intermediate (I-3e, middle panel), and linear intron (I, right panel) over time. Error bars represent standard errors of the mean (s.e.m.) calculated from $n = 3$ independent experiments. Source data are provided as a Source Data file.

C289U/C358U/G385A, aka the U-mutant), we monitored effects on splicing kinetics.

We found that the A-mutant—which can be protonated at position 358—splices at rates comparable to wild type, whereas the G and U mutants—which cannot be protonated at position 358—have splicing defects. Specifically, in the presence of near-physiological potassium and magnesium concentrations, the first splicing step of the G-mutant is ~12-fold slower and that of the U-mutant ~7-fold slower than in wild type. Moreover, the second splicing step of the G-mutant is ~48-fold slower and that of the U-mutant ~8-fold slower than in wild type (Fig. 1b, c and Supplementary Table 2). Most remarkably, both G and U mutants show accumulation of linear I-3e intermediate, which indicates that these mutants stall after the first step of splicing and have difficulty progressing into the second step (Fig. 1c, middle panel). These defects are comparable to those of other intron mutants designed to perturb the catalytic triad, such as ai5γ intron double mutants that carry G or U mutations at the nucleotide position analogous to O. iheyensis residue 358 and compensatory mutations of its corresponding Watson–Crick pair[33]. Finally, the splicing defects of our triple mutants are comparable to those of other O. iheyensis group II intron mutants designed to impair toggling, such as the C377G mutant reported in previous studies[16]. In this way, our enzymatic data connect defects in the transition between the two steps of splicing to specific active site mutations that prevent protonation on C358.

**The mutants are structurally intact but do not toggle**. To understand the splicing defects of the G and U mutants at the molecular level, we inserted the corresponding mutations into the previously described Oi5eD1-5 construct[16] and visualized the mutant active site by X-ray crystallography.

First, we determined crystal structures of the G and U mutants in the presence of potassium and magnesium at 3.4 and 3.6 Å resolution (Table 1). Both mutants have a folded structure similar to that of the post-hydrolytic state of the wild-type intron after the first step of splicing (PDB id.: 4FAR; root mean square deviation (RMSD)$_{4FAR-Gmutant}$ = 0.49 Å, RMSD$_{4FAR-Umutant}$ = 0.43 Å; Fig. 2a, d). Importantly, both mutant structures adopt the triple-helical configuration that corresponds with that of the wild-type intron structure (Fig. 2a and Supplementary Fig. 1d). The $F_o - F_c$ simulated-annealing electron density omit maps calculated by omitting the J2/3 residues and the catalytic metal cluster reveal strong electron density signal for the triple helix conformer, as in wild type (total peak height for the nucleobase of G288 = 8.9 σ and 6.7 σ for the G and U mutants, respectively; maximum peak height for the metals = 9.5 σ and 6.6 σ for the G and U mutants, respectively; Fig. 2b). Moreover, the $F_o - F_c$ maps calculated by omitting the first intron nucleotide (G1) show that the 5′-splice junction has undergone cleavage in both mutants during the crystallization process (Fig. 2c). In summary, the similarity of these mutant structures with that of wild type suggests that, despite some reductions in rate, the first step of splicing is structurally and mechanistically unaffected by the G and U mutations.

We then determined the crystal structures of the G and U mutants in the presence of sodium and magnesium at 3.2 and 3.3 Å resolution, respectively (Table 1). In this case, both mutants adopt overall structures similar to wild type (PDB id.: 4FAX; RMSD$_{4FAX-Gmutant}$ = 3.9 Å, RMSD$_{4FAX-Umutant}$ = 0.75 Å; Fig. 3a). However, the detailed architecture of the active site differs significantly from wild type under sodium conditions. For wild type, these conditions induce a rotation of the backbone in the J2/3 region, which breaks the triple helix structure and generates the so-called toggled conformation that is implicated in the transition between the first and the second step of splicing[16] (Fig. 3b, c). By contrast with wild type, the G and U mutants in sodium maintain the triple helix configuration, as revealed by the $F_o - F_c$ maps calculated by omitting the J2/3 residues and the catalytic metals (total peak height for the triple helix conformer of

**Table 1 Data collection and refinement statistics (molecular replacement).**

|  | G-mutant (Mg$^{2+}$-K$^+$) 6T3K | U-mutant (Mg$^{2+}$-K$^+$) 6T3R | G-mutant (Mg$^{2+}$-Na$^+$) 6T3N | U-mutant (Mg$^{2+}$-Na$^+$) 6T3S |
|---|---|---|---|---|
| Data collection |  |  |  |  |
| Space group | $P2_12_12_1$ | $P2_12_12_1$ | $P2_12_12_1$ | $P2_12_12_1$ |
| Cell dimensions |  |  |  |  |
| $a, b, c$ (Å) | 88.7, 95.4, 225.0 | 89.5, 95.1, 222.4 | 89.9, 95.3, 217.8 | 89.9, 95.2, 227.9 |
| $\alpha, \beta, \gamma$ (°) | 90, 90, 90 | 90, 90, 90 | 90, 90, 90 | 90, 90, 90 |
| Resolution (Å) | 48.5-3.44 (3.63-3.44)[a] | 44.5-3.57 (3.76-3.57)[a] | 32.1-3.22 (3.39-3.22)[a] | 49.5-3.28 (3.46-3.28)[a] |
| $R_{merge}$ | 5.3 (150.8)[a] | 13.0 (79.8)[a] | 6.9 (54.2)[a] | 7.4 (99.5)[a] |
| $I/\sigma I$ | 17.0 (1.6)[a] | 9.4 (2.5)[a] | 14.5 (2.4)[a] | 15.8 (2.5)[a] |
| Completeness (%) | 94.1 (95.0)[a] | 98.7 (99.3)[a] | 98.6 (99.7)[a] | 99.9 (99.8)[a] |
| Redundancy | 3.3 (3.2)[a] | 4.7 (4.8)[a] | 4.8 (5.0)[a] | 6.5 (6.5)[a] |
| Refinement |  |  |  |  |
| Resolution (Å) | 3.44 | 3.57 | 3.22 | 3.28 |
| No. reflections | 22183 | 22945 | 28966 | 29194 |
| $R_{work}/R_{free}$ | 21.2/25.7 | 21.2/25.0 | 18.5/24.2 | 19.8/22.9 |
| No. atoms |  |  |  |  |
| RNA | 8478 | 8450 | 8435 | 8390 |
| Ligand/ion | 101 | 107 | 61 | 63 |
| Water | 38 | 49 | 25 | 25 |
| $B$-factors |  |  |  |  |
| RNA | 129.18 | 122.57 | 141.54 | 132.64 |
| Ligand/ion | 84.00 | 101.51 | 143.21 | 96.32 |
| Water | 67.54 | 97.92 | 97.33 | 91.15 |
| R.m.s. deviations |  |  |  |  |
| Bond lengths (Å) | 0.009 | 0.002 | 0.009 | 0.008 |
| Bond angles (°) | 1.89 | 0.601 | 1.96 | 1.792 |

One single crystal was used to collect each data set, respectively.
[a]Values in parentheses are for the highest-resolution shell.

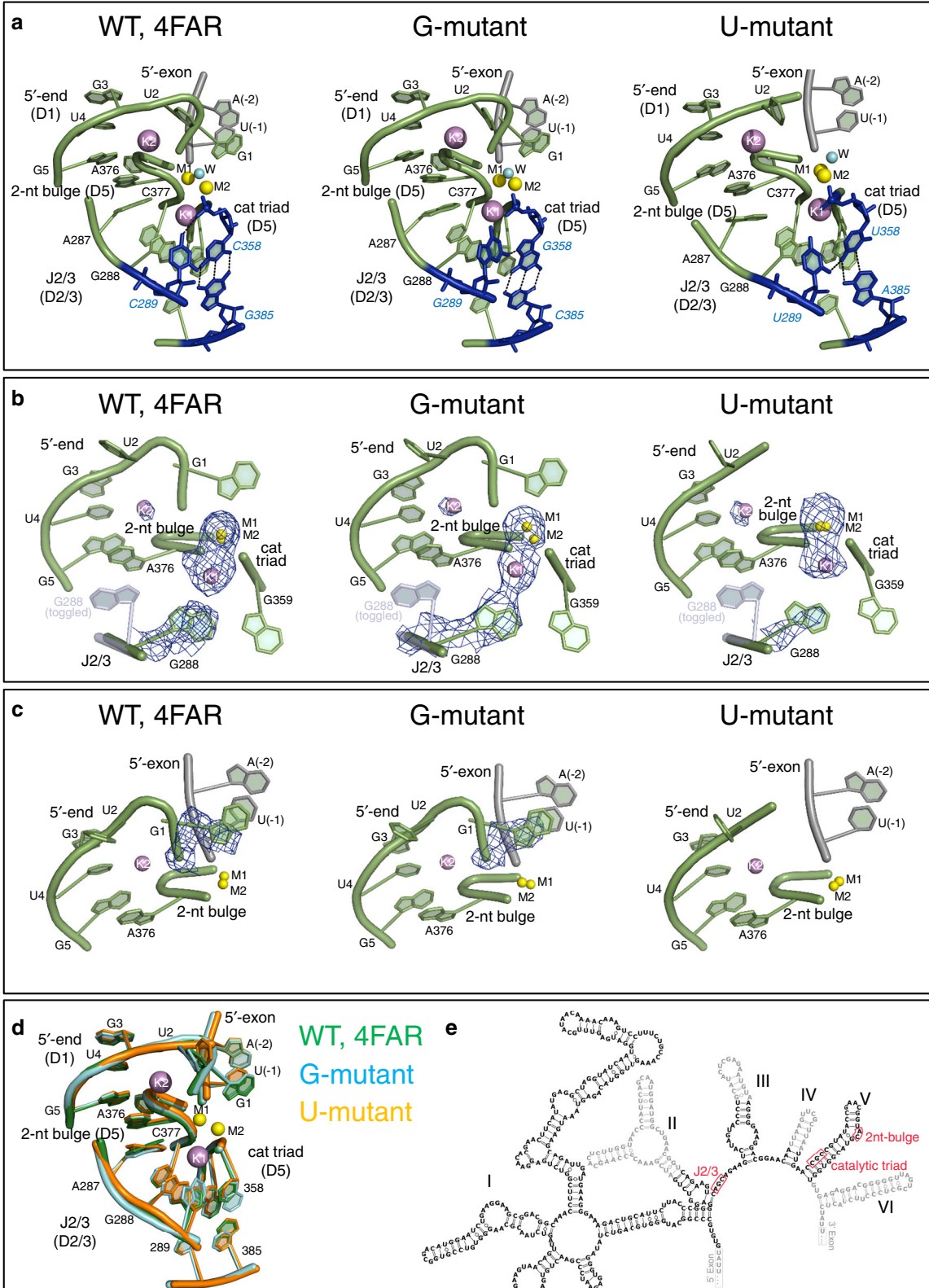

the G288 nucleobase = 7.5 σ in the G-mutant and = 9.3 σ in the U-mutant; Fig. 3b, c). Therefore, these structures show that the G and U mutants are unable to adopt the toggled conformation, which may explain their tendency to stall after the first step of splicing.

Taken together, the enzymatic and structural data suggest that C358 protonation and active site toggling facilitate the rearrangement of the intron active site between the two steps of splicing.

**Scission of the 5e disrupts the catalytic metal cluster.** To establish how C358 protonation and active site toggling are mechanistically connected, and to understand the chain of events that regulate active site rearrangement, we performed force-field-based MD simulations. We used a flexible nonbonded approach for the metal center (see "Methods" section), followed by comparative analyses of multiple systems built using the published

**Fig. 2 Crystal structures of the intron in potassium and magnesium. a** From left to right: Active site of wild type (WT; PDB id: 4FAR), G-mutant, and U-mutant in the state following 5′-exon hydrolysis. The intron is depicted as a cartoon representation in light green, the 5′-exon is in gray, and the mutated triplex is in blue (black dotted lines indicate H-bonds forming the triple helix). **b** $F_o − F_c$ simulated-annealing electron density omit-maps calculated by omitting J2/3 residues (nt 287–289) and the M1-M2-K1-K2 metal cluster from each of the structures depicted in (**a**). Positive electron density peaks within 3 Å from the omitted atoms are depicted as blue mesh at a contour level of 3 σ. The toggled conformation of G288 (from PDB id: 4FAX) is depicted as a semi-transparent light blue cartoon representation for reference. **c** $F_o − F_c$ simulated-annealing electron density omit-maps calculated by omitting G1 from each of the structures depicted in (**a**). Positive electron density peaks within 3 Å from the omitted atoms are depicted as blue mesh at a contour level of 3 σ. G1 is unresolved in the structure of the U-mutant. **d** Superposition of the active sites of the structures depicted in (**a**). **e** Secondary structure of the wild-type *O. iheyensis* group II intron with catalytic site elements highlighted by red boxes and nucleotides not present in the crystal structures in gray.

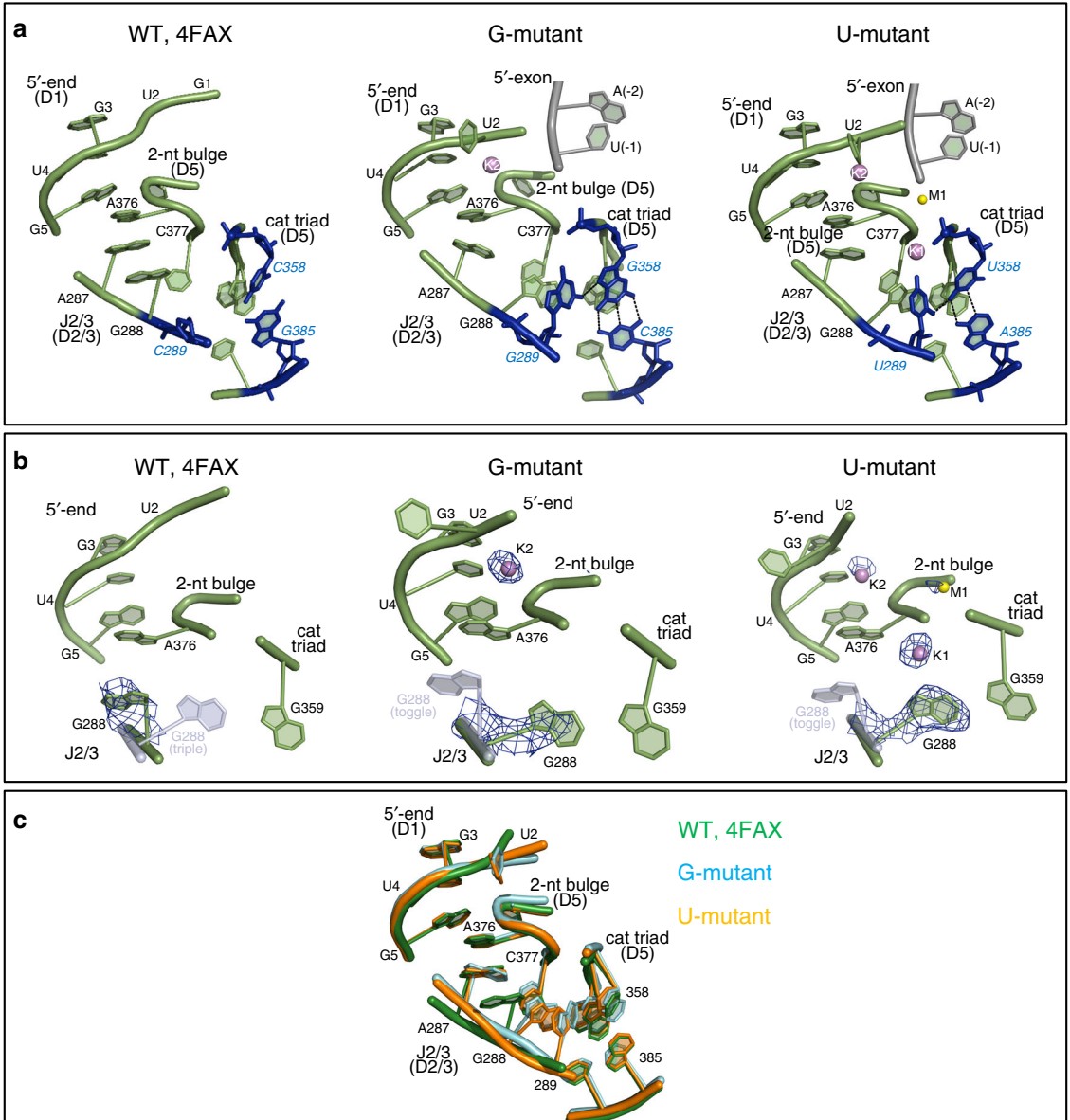

**Fig. 3 Crystal structures of the intron in sodium and magnesium. a** From left to right: Active site of wild type (WT; PDB id: 4FAX), G-mutant, and U-mutant. Active site elements are depicted as in Fig. 2a. **b** $F_o − F_c$ simulated-annealing electron density omit-maps calculated by omitting J2/3 residues (nt 287–289) and the M1–M2–K1–K2 metal cluster from each of the structures depicted in (**a**). Positive electron density peaks within 3 Å from the omitted atoms are depicted as blue mesh at a contour level of 3 σ. For the wild-type structure (left), the triple helix conformation of G288 is depicted as semi-transparent light blue cartoon representation, the M1–M2 metals as semi-transparent yellow spheres, and the K1–K2 metals as semi-transparent violet spheres (all using coordinates from PDB id: 4FAR). Analogously, the toggled conformation of G288 and unresolved metals of the cluster are depicted as semi-transparent representations for the G- and U-mutant (middle and right panels, using coordinates from PDB id: 4FAX for G288 and 4FAR for unresolved metals). **c** Superposition of the active sites of the structures depicted in (**a**).

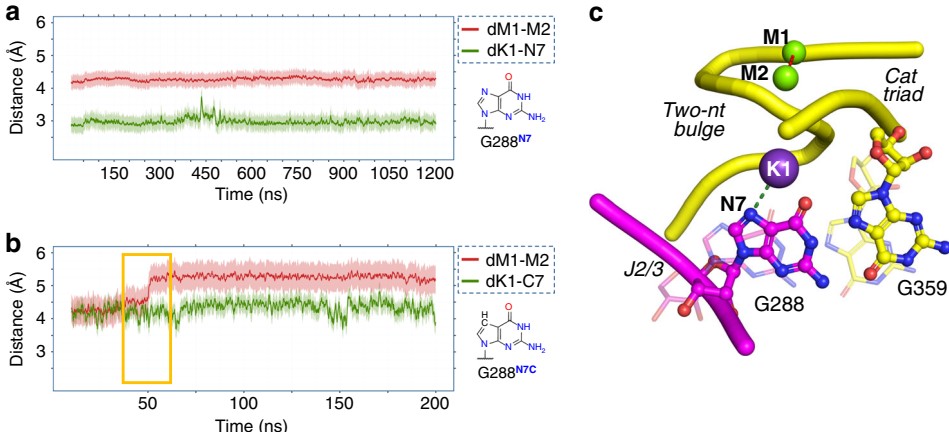

**Fig. 4 Importance of the K1 interaction with N7$^{G288}$. a** Changes during MD simulations of the pre-hydrolytic state for key structural descriptors $d_{K1-N7G288}$ (green trace) and $d_{M1-M2}$ (red trace). Shading around the traces indicates the standard deviation (s.d.) of the corresponding distance. The pre-hydrolytic state is stable and does not undergo any structural rearrangement, but it forms the K1–N7 interaction rapidly after the equilibration phase. **b** MD simulations of the N7-deaza state (structure of N7-deaza-G in the inset). The absence of the K1–N7 interaction alters the conformation of G288, eventually leading to active site misfolding ($d_{M1-M2} = 5.28 \pm 0.12$ Å, red trace). **c** Graphical representation of the K1–N7 interaction modeled from the simulations of the pre-hydrolytic state. The J2/3 junction (purple, ball and stick representation), M1 and M2 (green spheres), and the catalytic triad and two-nucleotide bulge (yellow, ball and stick representation, backbone as ribbon) are highlighted, together with the key descriptors reported in **a** and **b** (dotted lines represent $d_{K1-N7G288}$ and $d_{M1-M2}$). Disruption of K1–N7 causes G288 and G359 to move from their triple helix conformation (i.e. from the pre-hydrolytic state, solid colors) to a state in which the triple helix is broken (i.e. as simulated for the N7-deaza state, semi-transparent representation).

structures of the wild-type *O. iheyensis* group II intron captured at different stages of catalysis[16,20] and the structures of the G and U mutants. These structures represent the highest resolution crystallographic data available for group II introns and display the most detailed architecture of an intron active site, including all metals and first splicing step reactants[16,20].

We initially investigated the dynamics of the wild-type intron in the pre-hydrolytic state (PDB id: 4FAQ; two classical MD simulations for ~600 ns and ~1.2 µs, respectively). We observed that, shortly after equilibration (~25 ns), K1 shifted closer to the N7 atom of G288 (N7$^{G288}$), which was concomitant with the weakening of the K1 interaction with O5′$^{G359}$ observed in the crystal structure ($d_{K1-N7G288} = 2.98 \pm 0.27$ Å in the simulations, $d_{K1-N7G288} = 4.3$ Å in PDB id: 4FAQ, Fig. 4 and Supplementary Fig. 3). In both simulations, the system was structurally stable, especially nucleotides within the active site (domain D5 and junction J2/3). This was reflected in the average RMSD = $1.95 \pm 0.27$ Å (Supplementary Fig. 3) and the fact that catalytic triad residues maintained positions observed in the crystal structure ($d_{M1-M2}$ $4.24 \pm 0.04$ Å in the simulations, $d_{M1-M2} = 4.3$ Å in PDB id: 4FAQ). These simulations suggest that the pre-hydrolytic configuration does not have a tendency to undergo structural rearrangements.

We then investigated the dynamics of the wild-type intron after 5e hydrolysis, thus considering the post-hydrolytic state (PDB id: 4FAR) in protonated (three simulations, ~350 ns each) and non-protonated (six simulations, ~750 ns each) configurations (Supplementary Fig. 4). In these simulations, the overall structural fold was stably maintained, with an averaged RMSD of $4.72 \pm 0.65$ Å. As in the simulations of the pre-hydrolytic state, K1 shifted closer to the N7$^{G288}$ after equilibration (~25 ns, $d_{K1-N7G288} = 2.82 \pm 0.15$ Å in the simulations, $d_{K1-N7G288} = 4.4$ Å in PDB id: 4FAR). However, none of the post-hydrolytic systems were able to release the products of the first step of splicing. For example, the SP appears locked by the K2 ion in the proximity of the active site and the nucleobase of G1 remains stably coordinated to M1–M2 (Supplementary Fig. 4). These observations suggest that the post-hydrolytic crystal structures used in these simulations may represent an unproductive low energy configuration of the intron

that is not directly relevant to the pre-second step splicing configuration.

To address this issue, we modeled an active site state of the wild-type intron that would provide an improved starting point for simulations. We started with the structure of the pre-hydrolytic state (PDB id: 4FAQ), broke the scissile bond, and inverted the stereochemistry of the SP (further modeling details in "Methods" section and in Supplementary Fig. 5). This state represents the intron immediately after the first step of splicing, where the SP has just been cleaved but is still coordinated by M1 and M2 (Supplementary Fig. 5). Also for this 'cleaved' state, we simulated both protonated and non-protonated forms of C358 (two simulations per system, ~600 ns per simulation). In all cases, the system showed considerable stability, with an overall RMSD of $4.61 \pm 0.81$ Å. During these simulations, the K1–N7$^{G288}$ interaction was formed and initially preserved. Moreover, the SP was not sequestered by K2 outside the active site. In other words, the distance between the SP and M2 was constantly maintained at $d_{SP-M2} = 3.23 \pm 0.10$ Å (Supplementary Fig. 5). Intriguingly, in the protonated state, after ~20 ns of simulation, a water molecule bridged O6$^{G288}$ and M2, such that these two atoms became closer to each other ($d_{M2-O6} = 5.71$ Å in PDB id: 4FAQ; $d_{M2-O6} = 4.75 \pm 0.23$ Å in the simulations; Fig. 5 and Supplementary Fig. 5). Concomitantly, the value of $d_{M1-M2}$ increased from 4.31 to $5.05 \pm 0.13$ Å (Fig. 5 and Supplementary Fig. 5). Importantly, at this point, the coordination shell of K1 was perturbed, and the K1–N7$^{G288}$ interaction broke, leading to the spontaneous release of K1 from the active site into the bulk solvent after just additional ~30 ns (Fig. 5 and Supplementary Fig. 5). Notably, these events occurred also in the non-protonated state, although less promptly. In this case, the initial conformational changes occurred after ~200 ns, with K1 released soon after, at ~250 ns. Interestingly, in all cases, the conformational ensemble of the active site after the release of K1 differed from the characteristic triple helix configuration. To specifically monitor the triple helix geometry, we used the following two geometrical parameters: (1) the distance between the O2 atom of C289 (J2/3) and the N4 atom of G358 (D5, catalytic triad) ($d_{289-358}$), which adopts values ≤ 3 Å in the triple helix configuration and > 3 Å

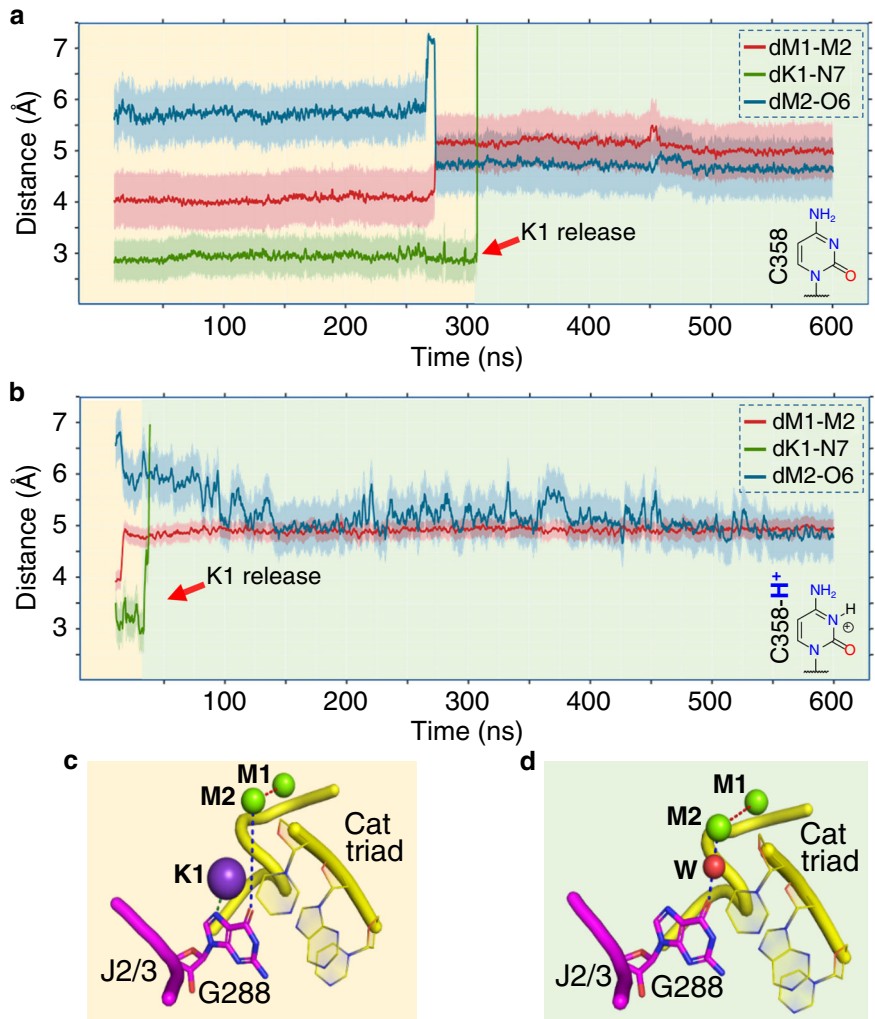

**Fig. 5 Protonation of N3$^{C358}$ favors K1 release.** Evolution of $d_{M1-M2}$ (red trace), $d_{K1-N7}$ (green trace), and $d_{M2-O6}$ (blue trace) over the course of MD simulations of: **a** the cleaved state, and **b** the cleaved-H$^+$ state. Shading around the traces indicates the s.d. of the corresponding distance Schematic structures of non-protonated and protonated cytidine groups are depicted in the bottom right of each panel. K1 is released after ~300 ns and ~40 ns from the two states, respectively (red arrow). **c** Structure of the active site at the beginning of the simulations after the formation of K1–N7$^{G288}$. **d** Structure of the active site at the end of the simulations, after K1 release. K1 is released after the solvation of the active site (water molecule depicted as a red sphere, W).

when the triple helix is disrupted; and (2) the angle $\alpha$ between the nucleobases plains of C358 and its Watson–Crick pair G385, which adopts values $\leq 0.35$ rad in the triple helix configuration and $>0.35$ rad when the triple helix is disrupted (Supplementary Fig. 6). Indeed, $d_{289-358} = 2.7$ Å and $\alpha = 0.17$ rad in the crystallized pre-hydrolytic state (PDB id: 4FAQ), which harbors K1 and adopts the triple helix configuration. Notably, though, after K1 release in our MD simulations, $d_{289-358}$ reached average values of ~4.88 ± 1.05 Å and $\alpha$ reached average values of ~0.63 ± 0.14 rad in the protonated state (~3.07 ± 0.14 Å and ~0.47 ± 0.10 rad in the non-protonated state, respectively), suggesting that the triple helix is destabilized and the active site may toggle under these conditions (Supplementary Fig. 6).

Finally, we also simulated the crystallized G and U mutants in the cleaved and post-hydrolytic states (eight simulations, ~600 ns each; Supplementary Figs. 7 and 8). We noted that the K1–N7$^{G288}$ interaction was not stably formed in the mutants, preventing K1 release (Supplementary Figs. 7 and 8). Thus, the triple helix was stabilized in its crystallographic conformation. For example, in the simulations, $d_{289-358} = 2.67 \pm 0.24$ Å and $\alpha = 0.19 \pm 0.11$ rad for the G-mutant ($\alpha = 0.15$ rad in PDB id 6T3K)

and $d_{289-358} = 1.95 \pm 0.23$ Å and $\alpha = 0.33 \pm 0.13$ rad for the U-mutant ($\alpha = 0.31$ rad in PDB ID=6T3R) (Supplementary Figs. 7 and 8). These simulations suggest that the G and U mutants are unlikely to toggle in their cleaved form.

Taken together, these data suggest that K1 is stably bound to the active site in the pre-hydrolytic state of the wild-type intron, but is spontaneously released from the active site immediately after 5e hydrolysis. The release of K1 breaks the catalytic triple helix, and the intron begins sampling the toggled conformation. Such a rearrangement is significantly favored by protonation of N3$^{C358}$, and it does not occur in the G and U mutants, which cannot be protonated.

**The K1–N7$^{G288}$ interaction stabilizes the intron active site.** Interestingly, in the simulations of the wild-type intron described above, but not in the simulations of the mutants, K1 establishes a stable interaction with N7$^{G288}$ within a very short time after equilibration (Figs. 4, 5 and Supplementary Figs. 4–8). Moreover, simulations of the cleaved state immediately after 5e hydrolysis showed that interaction with N7$^{G288}$ is a necessary step for

releasing K1 from the active site (Fig. 5 and Supplementary Fig. 5). Importantly, an N7-deaza mutation at position G288 was shown to impair the first step of splicing[34]. These observations suggest that the K1–N7$^{G288}$ interaction may be structurally and functionally important for splicing.

To test this hypothesis, we modeled the N7-deaza mutation at G288 in the pre-hydrolytic state (PDB id: 4FAQ), and we tested the importance of the K1–N7$^{G288}$ interaction for the proper folding of the active site. Three classical MD simulations of these in silico mutants (~200 ns each) showed that the loss of the K1–N7$^{G288}$ interaction irreversibly destabilized the triple helix, causing separation of M1–M2 (averaged $d_{M1-M2} = 5.28 \pm 0.12$ Å, Fig. 4b) and eventually leading to the unfolding of the active site.

These data suggest that the K1–N7$^{G288}$ interaction plays a crucial role in preventing premature release of K1 and consequent disruption of the triple helix.

**Toggling energetics agree with catalytic rate constants**. To appropriately sample and semi-quantitively evaluate the energetics associated with intron toggling, we used path-metadynamics (MtD)[35]. We performed enhanced sampling MtD simulations starting from either the cleaved protonated or non-protonated wild-type models and terminating at the toggled state (referred to as the $cH^+ \to T$ and the $c \to T$ transitions, respectively; see details in "Methods" section). The reference path involves exclusively the J2/3 junction, which rearranges as defined from structural data[16], and employs two collective variables that trace (1) the progress of the system along the reference path (variable $S$), and (2) the distance of the sampled conformations from the reference path (variable $Z$). In this way, MtD simulations sample the conformational space to find the lowest energy path for the conformational change under investigation. Notably, the non-bonded metal cluster M1–M2–K1–K2 and its extended coordination shell at the catalytic site can freely explore conformational space during these simulations.

Mechanistically, in simulations where C358 was protonated, the system first sampled a large, deep free energy minimum that contained multiple isoenergetic conformational states. While A287 freely explored the conformational space, C358 protonation disrupted the canonical WC base pairing with G385, leading to C358 rotation ($d_{289-358} = 5.85 \pm 1.94$ Å and $\alpha = 0.34 \pm 0.08$ rad, state A, Fig. 6). This spontaneous rearrangement promoted hydration of the K1-binding site, with consequent prompt release of K1 to the bulk solvent, disruption of the hydrogen-bond contacts between C358 and C289, and further separation of these two residues ($d_{289-358} = 10.33 \pm 1.75$ Å and $\alpha = 1.24 \pm 0.18$ rad, state B, Fig. 6). In this conformation, the flexibility of J2/3 nucleotides was enhanced, allowing G288 and C289 to rotate out of the active site and to stack with A287, thus enabling the disruption of the C377–C360 base pair (Toggled state, Fig. 6). The computed energetic barrier for this overall transition ($\Delta G^{\ddagger}_{cH^+-T}$) was ~20 kcal mol$^{-1}$, while the final metastable toggled state had a value of about +5 kcal mol$^{-1}$ relative to the triple helix conformer (Fig. 6).

Nucleotides within J2/3 also rearranged in the non-protonated configuration, albeit with different dynamics and higher energy barriers. Indeed, with the spontaneous rotation of A287, the intron rearranged into the first intermediate state ($d_{289-358} = 2.97 \pm 0.21$ Å and $\alpha = 0.17 \pm 0.08$ rad, state A′, Supplementary Fig. 9, which is the lowest free energy minimum), in which K1 is more exposed to the bulk water. K1 is then released simultaneously to the partial rotation of G288. This led to the disruption of the triple helix and the formation of a second intermediate state ($d_{289-358} = 7.97 \pm 1.19$ Å and $\alpha = 0.31 \pm 0.16$ rad, state B′, Supplementary Fig. 9). Finally, the stacking of A287

and C289 with G288, together with the rotation of C377, completed the conformational rearrangement and formed the final toggled state (Supplementary Fig. 9). The computed free energy barrier for this transition ($\Delta G^{\ddagger}_{c-T}$) was ~25 kcal mol$^{-1}$, which is therefore less favorable than that of the protonated intron ($\Delta G^{\ddagger}_{cH^+-T}$ ~20 kcal mol$^{-1}$). Importantly, the final metastable toggled state had a value of about +5 kcal mol$^{-1}$ relative to lowest free energy minimum (state A′, Supplementary Fig. 9). These computed activation barriers are a good match with empirical values calculated using the experimental splicing rate constants inserted into the Eyring–Polanyi equation[36,37] ($k_1 = 0.031 \pm 0.003$ min$^{-1}$ → $\Delta G^{\ddagger}_1 = 22.8$ kcal mol$^{-1}$; $k_2 = 0.026 \pm 0.003$ min$^{-1}$ → $\Delta G^{\ddagger}_2 = 22.9$ kcal mol$^{-1}$). This result corroborates our proposed toggling mechanism, indicating that conformational rearrangements of the intron active site between the catalytically active triple helix configuration and the toggled structure captured crystallographically[16] are compatible with catalysis.

## Discussion

By combining structural, enzymatic, and computational methods, we have elucidated the molecular mechanism for the transition between the two steps of group II intron splicing and we have described the dynamic behavior of the intron active site as it moves through the splicing process (Fig. 7 and Supplementary Movie 1).

In the pre-hydrolytic state, the group II intron adopts the triple helix conformation, which coordinates the heteronuclear metal cluster M1–M2–K1–K2[16,19]. M2 and the phosphate backbone of C358 deprotonate the reaction nucleophile for catalyzing the scission of the 5′-splice-site. At this stage, the proton released into bulk solvent by the reaction nucleophile[24] is transferred to the N3 atom on the C358 nucleobase, either via specific proton transfer pathways, as previously proposed[24] (an illustration of one possible transfer pathway is reported in Supplementary Fig. 2a) or via simple diffusion through the solvent. Independent on the exact proton transfer mechanism, hybrid quantum-classical simulations suggest that C358 remains stably protonated on N3, never exchanging its proton with surrounding water in the quantum region (Supplementary Fig. 2b, c). Indeed, it is remarkable that position 358 is often occupied by an adenosine (which is readily protonated) in the majority of group II introns and in the spliceosome, but it never varies to G or U (which are nucleobases that cannot be protonated). C358 protonation thus emerges as a previously unrecognized event that stimulates the progression of the intron toward the second step of splicing. Indeed, when we experimentally replaced C358 with adenosine, splicing was unaffected. But when we replaced C358 with either a G or a U, the mutated intron accumulated linear I-3e intermediate, indicating a defect in the progression onto the second step of splicing. Notably, and in line with the MD simulations of protonated and non-protonated wild-type intron in the cleaved post-hydrolytic state (see below), splicing is not completely inhibited in the mutants, suggesting that protonation accelerates splicing but it is not essential. These differences in kinetics likely constitute a phenotypic advantage for the intron, which has preserved protonatable residues at position 358 throughout evolution. When position 358 is occupied by a G or U residue, steric or electrostatic perturbations may also contribute to the observed splicing defects. The extent of such perturbations may be qualitatively inferred from the behavior of the A-mutant, which—despite being protonatable at position 358—contains a bulkier purine substitute. Remarkably, splicing defects of the A-mutant are minimal (approximately twofold, Fig. 1b,c and Supplementary Table 2) and crucially, they do not lead to accumulation of an I-3e

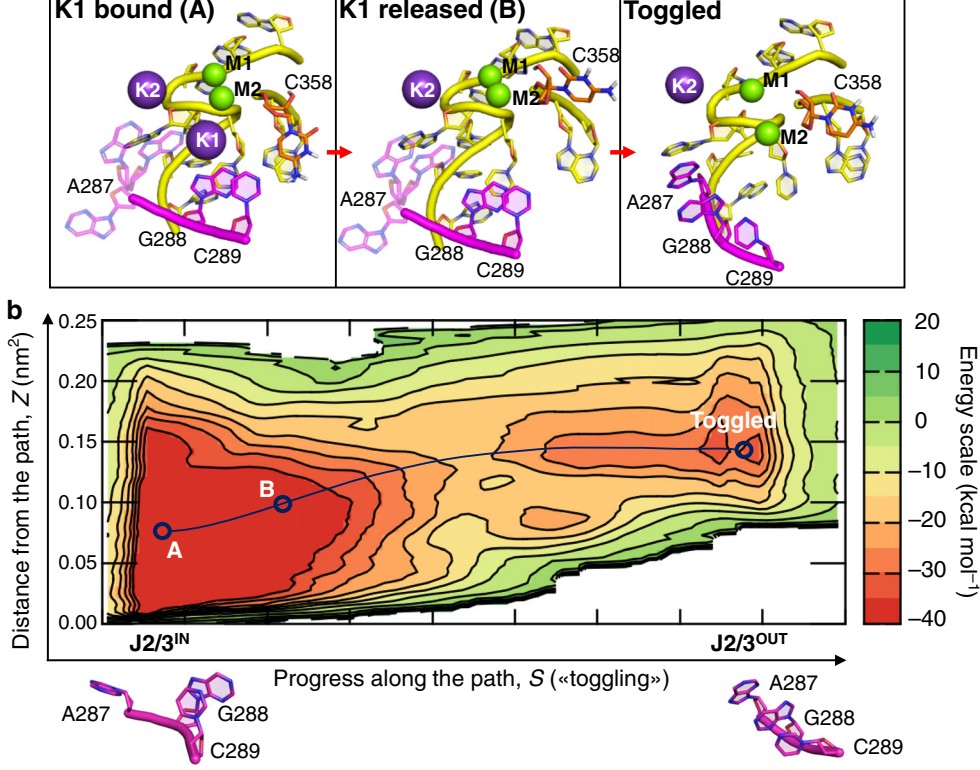

**Fig. 6 Energetics associated with intron toggling in the protonated state. a** Structural architecture of the active site for the intermediate A, intermediate B, and toggled states identified by the MtD simulation reported in (**b**). **b** Path MtD free energy landscape of the cleaved-H$^+$ state. The intermediate A, intermediate B, and toggled states are indicated along the MtD trajectory (dark blue dotted line). The energy scale is indicated in kcal mol$^{-1}$ on the right. The conformations of the J2/3 junction in state A and in the toggled state are represented at the bottom of the figure.

intermediate (Fig. 1b, c and Supplementary Table 2), suggesting that the defects of the G and U mutants predominantly derive from their inability to become protonated at position 358. The crystal structures of these mutants additionally confirm that these constructs preserve triple helix architecture, so any perturbation of their active site must be minimal. Finally, the crystallographic data suggest that the G and U mutants do not sample the toggled conformation, thus impairing a critical rearrangement of the intron active site between the two steps of splicing.

MD simulations enabled us to dissect the precise sequence of events that lead from 5′-splice site cleavage to toggling. Most importantly, we observed that 5e hydrolysis induces a spontaneous and prompt release of K1 from the active site, as previously hypothesized[16]. This key event, which happens only in the post-hydrolytic but not in the pre-hydrolytic state, is much favored by protonation of C358, which induces fast K1 release (just after ~50 ns when protonated, as compared to ~250 ns in the non-protonated state, with an energetic barrier of $\Delta G^{\ddagger}_{cH^+-T} \sim 20$ kcal mol$^{-1}$ and $\Delta G^{\ddagger}_{c-T} \sim 25$ kcal mol$^{-1}$, respectively). These observations reveal that K1 is a highly dynamic ion, despite its tight coordination to nearly all active site residues in the catalytic triple helix configuration, and it plays an extremely crucial role during splicing. The interaction of K1 with N7$^{G288}$, in the J2/3 junction, is particularly important for stabilizing the intron active site in the catalytically competent configuration and for controlling the binding and release of K1 within the active site along the splicing cycle. These results explain why N7$^{G288}$-deaza mutants are defective in splicing[34].

As a result of K1 release, the triple helix conformation becomes unstable. Under such circumstances, G288 toggles out of the active site, undergoing backbone rotations that expose the

Watson–Crick face of guanosine to functional groups in D3 and the Hoogsteen face to a cavity that is likely occupied by D6 and by 3′-splice junction residues[9,11,13,16]. In this conformation, G288 is thus optimally placed to promote key interactions that facilitate the second step of splicing (see below). MtD simulations show that the energy required for such conformational toggling is compatible with catalytic rate constants. Importantly, mutants that are defective for toggling, either because they cannot be protonated or because their triple helix is stable even under conditions where the wild-type toggles (i.e. the G and U mutants described here, and the C377G mutant studied previously[16]), fail to progress onto the second step of splicing.

Toggling of the J2/3 junction and progression to the second step of splicing is also likely to involve A287 (nucleotide γ). In our simulations, A287 establishes a canonical WC interaction with the second nucleobase of the intron (U2; $d_{U2-A287} = 2.13 \pm 0.32$ Å; Supplementary Fig. 10), which was maintained as long as K1 remained in the active site, but which was broken when K1 left the active site and the intron toggled. Such findings suggest that G288 toggling may be needed to release A287 from U2. This process would ensure the formation of the essential γ–γ′ interaction, in which A287 pairs with its partner nucleotide γ′ in D6 during the second step of splicing[9,34,38]. After recruiting D6 via A287, the toggled intron would then re-establish the catalytic triple helix conformation by reverse toggling, explaining how the first and second steps of splicing are mechanistically connected. Based on the simulations, the toggled state is ~5 kcal mol$^{-1}$ higher in free energy compared to the triple helix state, suggesting that reverse toggling is energetically inexpensive. It is therefore tempting to speculate that protonation and toggling also occur at the end of the second step, which might favor the

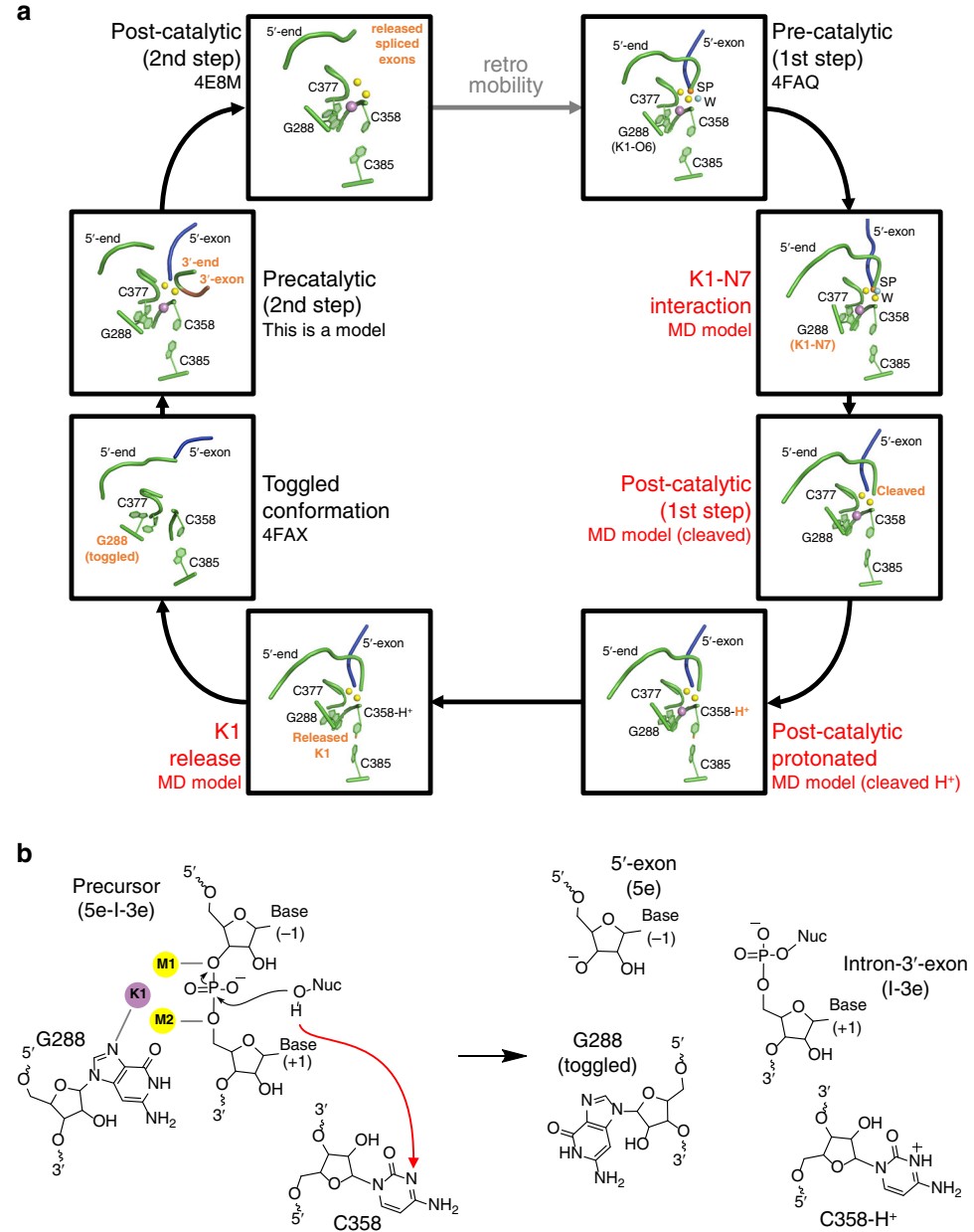

**Fig. 7 Revised group II intron splicing cycle. a** Snapshots of the intron active site derived from the crystal structures and the MD simulations. In the pre-catalytic state, which corresponds to the pre-hydrolytic state of the first step of splicing, a water molecule (light blue sphere, W) or the 2′-OH of a bulged adenosine are poised for nucleophilic attack on the scissile phosphate (orange sphere, SP) (PDB id. 4FAQ). In this state, the intron established the K1–N7$^{G288}$ interaction, and the nucleophile cleaves the intron-5en junction. Immediately after hydrolytic cleavage, the proton released by the nucleophile initially to the bulk solvent is shuttled to C358 in the catalytic triad. C358 protonation favors release of the metal ions cluster and toggling of J2/3 junction (PDB id. 4FAX). This conformational rearrangement likely prompts the release of the products of the first step of splicing, subsequent rearrangement of D6, and reverse toggling of J2/3 to reconstitute the active site and align second step reactants. Finally, cleavage of intron-3e junction leads to the release of free linear intron (PDB id. 4E8M) and spliced exons. The free intron is still an active ribozyme, which can retrotranspose into target genomic DNA and re-initiate a new splicing cycle (dashed gray arrow). Relevant intron motifs are shown as cartoon representations in green. The 5e is in blue, the 3e in brown. K1 is shown as a violet sphere, M1 and M2 as yellow spheres. K2 is not shown for clarity. Intron states described in this work are labeled red, and differences between consecutive panels are indicated as bold orange labels. **b** Sketches of the intron active sites corresponding to the precatalytic state (with K1–N7 interaction, left) and to the K1 release and toggling (right) drawn in the same style as in Fig. 1a. For clarity, only the 5′-splice site is represented. The red dotted arrow indicates the proton transfer pathway from the nucleophile (Nuc) to N3$^{C358}$.

release of the splicing product, and reduce the chances of spliced exons reopening[22,23]. In either case, these processes may be further facilitated by participation of the intron-encoded maturase protein.

The idea of a rearrangement involving J2/3 and formation of a transiently inactive intermediate is compatible with the

mechanism of splicing via the branching pathway. Indeed, biochemical data and recent crystal structures of lariat introns show that the hydrolytic and transesterification pathways occur at the same active site, involve positioning of the reaction nucleophile (the proton donor) in the exact same structural position compared to nucleotide 358, and follow the same reaction

chemistry[9,11,13,20,21]. Indeed, G- and U-mutations at the 358-equivalent position of the lariat-forming ai5γ intron from *Saccharomyces cerevisiae* (A816G and A816U) display splicing defects similar to our G and U mutants[33]. Moreover, a protonation-dependent structural rearrangement mechanism is strongly supported by functional data obtained on the spliceosome, which is evolutionarily and chemically analogous to the group II intron[1,2]. In the spliceosome, the last G (G52 in yeast) of the conserved ACAGAGA box in U6 snRNA corresponds to intron G28831. This residue is in close proximity to the branch site[39], it interacts with the 5′-splice site, and it undergoes a rearrangement between the splicing steps[40] in a process that is modulated by protein subunits (i.e. Prp8, Prp16)[41,42] and potassium ions[43,44]. Such reorganization of G52 facilitates the release of the 5′-end of spliceosomal introns from the active site after the first splicing step, while also favoring the recruitment of the 3′-splice junction into the active site for the second step of splicing[40]. These rearrangements could be induced by protonation of nucleotides of the U6 ISL, which are analogous to the group II intron two-nucleotide bulge and catalytic triad because their protonation antagonizes binding of the catalytic metal ions to the spliceosome and induces a transient base-flipping conformational change[29,30]. Furthermore, G52 mutations in the spliceosome have an inhibitory effect on the second step of splicing[45], similar to the effects we described for G288 in the group II intron in this and in previous work[22,46]. Finally, during the splicing cycle, the spliceosome adopts transiently inactive states, possibly similar to the group II intron inactive toggled conformation, to avoid processing non-ideal pre-mRNA substrates[47]. In the light of these structural and functional analogies between the intron and the spliceosome, it seems plausible that conformational toggling and dynamics of catalytic metal ions in the active site may regulate spliceosomal activation, too.

In summary, through the integration of four X-ray structures of active site mutants and in vitro splicing assays with multi-microsecond classical molecular simulations and free energy calculations, we have elucidated the dynamical behavior and determined the functional role of structural rearrangements within the group II intron active site, showing how they contribute to the mechanism of RNA splicing. We have determined that critical dynamic processes are triggered by protonation of a highly-conserved catalytic residue, thereby promoting the transition between the first and the second steps of splicing. Importantly, the resulting mechanism explains the apparent paradox of how and why a tightly bound metal ion cluster can be broken and reformed during the catalytic cycle, thereby promoting a directional sequence of coordinated chemical reactions. These findings may help in future engineering of complex, RNA-based enzymes for use as biotechnological tools and gene-specific therapeutics[4,48].

## Methods

**Cloning and mutagenesis**. The constructs of *O. iheyensis* group II intron used in this work are the pOiA wild type and the OiD1-5 crystallization constructs of the *O. iheyensis* group II intron[16]. All mutagenesis experiments were performed using the PfuUltra II Hotstart PCR Master Mix (Agilent). The restriction enzymes *ClaI* and *BamHI* used for template linearization were purchased from NEB. All constructs were confirmed by DNA sequencing (W. M. Keck Foundation Biotechnology Resource Laboratory, Yale University, and Eurofins).

**In vitro transcription and purification**. Following restriction with the appropriate endonucleases at 37 °C overnight, the intron was transcribed in vitro using T7 polymerase[16]. For crystallization purposes[17,18], it was then purified under non-denaturing conditions[49], re-buffered, and concentrated to 80 μM in 10 mM MgCl₂ and 5 mM sodium cacodylate pH 6.5. For splicing studies, the intron was radio-labeled during transcription, purified in a denatured state[16], and subsequently refolded.

**Splicing assays**. Purified radiolabeled intron precursor was refolded by denaturation at 95 °C for 1 min in the presence of 40 mM Na-MOPS pH 7.5, and cooled at room temperature for 2 min. Subsequently, the appropriate monovalent ions were added to a final concentration of 150 mM. Finally, MgCl₂ was added to a final concentration of 5 mM to start the splicing reaction. The refolded precursor samples were incubated at 37 °C. One microliter aliquots of the splicing reactions taken at specific time points were quenched by the addition of 20 μL gel loading solution containing urea and chilled on ice. The samples were analyzed onto a denaturing 5% (w/v) polyacrylamide gel. The kinetic rate constants were calculated using the Prism 6 package (GraphPad Software).

**Crystallization**. The natively purified intron was mixed with a 0.5 mM spermine solution in 10 mM MgCl₂ and 5 mM sodium cacodylate pH 6.5, and with the crystallization buffer in a 1:1:1 volume ratio[16]. Crystals were grown at 30 °C by the hanging drop vapor diffusion method using 2 μL sample drops and 300 μL crystallization solution in a sealed chamber (EasyXtal 15-Well Tool, Qiagen). Crystals were harvested after 2–3 weeks. Crystals were cryo-protected in a solution containing the corresponding crystallization buffers supplemented with 25% EG and immediately flash frozen in liquid nitrogen. The crystallization solutions used to solve the structures of the excised intron presented in this work were composed of: (1) 50 mM Na-HEPES pH 7.0, 100 mM magnesium acetate, 150 mM potassium chloride, 10 mM lithium chloride, 4% PEG 8000 for the G-mutant in potassium and magnesium (PDB id: 6T3K), (2) 50 mM Na-HEPES pH 7.0, 100 mM magnesium acetate, 150 mM potassium chloride, 10 mM lithium chloride, 4% PEG 8000 for the U-mutant in potassium and magnesium (PDB id: 6T3R), (3) 50 mM Na-HEPES pH 7.0, 100 mM magnesium acetate, 150 mM sodium chloride, 4% PEG 8000 for the G-mutant in sodium and magnesium (PDB id: 6T3N), and (4) 50 mM Na-HEPES pH 7.0, 100 mM magnesium acetate, 150 mM sodium chloride, 4% PEG 8000 for the U-mutant in sodium and magnesium (PDB id: 6T3S).

**Structure determination**. Diffraction data were collected with an X-ray beam wavelength of 0.979 Å and at a temperature of 100 K at beamlines 24ID-C and E (NE-CAT) at the Argonne Photon Source (APS), Argonne, IL, and processed with the Rapid Automated Processing of Data (RAPD) software package (https://rapd.nec.aps.anl.gov/rapd/) and with the XDS suite[50]. The structures were solved by molecular replacement using Phaser in CCP4[51] and the RNA coordinates of PDB entry 4FAR (without solvent atoms) as the initial model[16–18]. The models were improved automatically in Phenix[52] and Refmac5[51], and manually in Coot[53], and finally evaluated by MolProbity[54]. The figures depicting the structures were drawn using PyMOL[55]. Stereo images of selected regions of the electron density are reported in Supplementary Fig. 11.

**pKₐ calculations**. We used continuum electrostatics calculations based on the nonlinear Poisson–Boltzmann equation to estimate the $pK_A$ of C358 in the prehydrolytic state (PDB id. 4FAQ) and in the toggled state (PDB id. 4FAU). Calculations were performed with DelPhiPKa[56], using a pH range from 0 to 14 with a pH interval of 0.5, a dielectric constant for RNA $\varepsilon_{RNA} = 4$, and a dielectric constant for solvent $\varepsilon_{solvent} = 80$. Metals were not considered in the calculations.

**Structural models for MD simulations**. We have used ten systems for MD simulations: (1) The pre-hydrolytic state, a wild-type system modeled on PDB id: 4FAQ;[16] (2) The N7-deaza state, a pre-hydrolytic state in which N7$^{G288}$ was replaced by a carbon atom; (3) The cleaved state, a pre-hydrolytic state in which the phosphodiester bond between the intron and the 5e was broken introducing an oxygen atom and inverting the stereochemical configuration of the SP, and in which Ca²⁺ ions were replaced with Mg²⁺ ions; (4) The cleaved-H⁺ state, a cleaved state protonated on N3$^{C358}$; (5) The post-hydrolytic state, a wild-type system modeled on PDB id: 4FAR;[16] (6) The post-hydrolytic H⁺ state, a post-hydrolytic state protonated at N3$^{C358}$; (7) The post-hydrolytic G-mutant, modeled on the structure of the G-mutant in potassium and magnesium; (8) The cleaved G-mutant, a cleaved state carrying the C289G/C358G/G385C triple mutations; (9) The post-hydrolytic U-mutant, modeled on the structure of the G-mutant in potassium and magnesium; (10) The cleaved U-mutant, a cleaved state carrying the C289U/C358U/G385A triple mutations. Each system was hydrated with a 12-Å layer of TIP3P[57] water molecules, and the ions concentration was set to the same used for crystallization[16]. All the crystallized ions and water molecules were considered for model building. The final models are thus enclosed in a box of ~145·125·144 Å³, containing ~220,000 water molecules, resulting in a total of ~250,000 atoms for each system.

**MD simulation set up**. The AMBER-ff12SB (ff99 + bsc0 + χOL3)[58] was used for the parametrization of the RNA. Nucleotide G288 in the N7-deaza model, nucleotide C358 in the cleaved-H⁺ and post-hydrolytic H⁺ models, and both 5′- and 3′- terminal nucleotides in all models were parametrized with the general amber force field (i.e. GAFF)[59], and their atomic charges were derived with RESP procedure[60]. We used the Joung–Cheatham parameters[61] for the monovalent metal ions, while the divalent metal ions were parametrized according to Li et al.[62]. In the simulations, we have used ionic concentrations of 100 mM for magnesium ions and 150 mM for potassium ions, in line with the crystallization conditions of the intron

(see above). The two catalytic metal ions were modeled using a flexible nonbonded approach based on the atoms in molecules partitioning scheme[63–65]. All MD simulations were performed with Gromacs 5.1.4[66]. The integration time step was set to 2 fs, while the length of all covalent bonds was set with the P-LINCS algorithm[67]. A temperature of 310 K was imposed using a velocity-rescaling thermostat[68] with a relaxation time $\tau = 0.1$ ps, while pressure control was achieved with Parrinello–Rahman barostat[69] at reference pressure of 1 atm with $\tau = 2$ ps. Periodic boundary conditions in the three directions of Cartesian space were applied. A particle mesh Ewald method, with a Fourier grid spacing of 1.6 Å, was used to treat long-range electrostatics. All the systems were subjected to the same simulation protocol. To relax the water molecule and the ions, energy minimization was carried out. At this stage, active core ions (M1, M2, K1, K2, K4[19]) along with the RNA backbone were kept fixed with harmonic positional restraints of 500 kcal/mol Å². Subsequently, the systems were heated up from 0 to 310 K with an NVT simulation of ~1 ns with the same positional restraints used in the energy minimization. A second NVT of ~1 ns was then performed at a fixed temperature (310 K), halving the positional restraints. In addition, ~1 ns of NPT simulation was performed with 100 kcal/mol Å² residual restraints on the backbone and the core ions to allow partial backbone relaxation. Finally, different production runs were performed in the NPT ensemble for each system. We collected overall more than 15 µs of MD trajectories, specifically: (1) ~1.8 µs for the pre-hydrolytic system, two replicas; (2) ~600 ns for the N7-deaza system, three replicas; (3) ~1.2 µs for the cleaved system, two replicas; (4) ~1.2 µs for the cleaved-H⁺ system, two replicas; (5) ~4.5 µs for the post-hydrolytic system, six replicas; (6) ~1 µs for the post-hydrolytic H⁺ state, a post-hydrolytic state protonated at N3$^{C358}$, three replicas; (7) ~1.2 µs for the post-hydrolytic G-mutant, modeled on the structure of the G-mutant in potassium and magnesium, two replicas; (8) ~1.2 µs for the cleaved G-mutant, a cleaved state carrying the C289G/C358G/G385C triple mutations, two replicas; (9) ~1.2 µs for the post-hydrolytic U-mutant, modeled on the structure of the G-mutant in potassium and magnesium, 2 replicas; (10) 1.2 µs for the cleaved U-mutant, a cleaved state carrying the C289U/C358U/G385A triple mutations, two replicas. For each system, statistics were collected after the systems reached the equilibration (i.e., stabilization of the RMSD of the nucleic acid backbone), thus discarding the first 25 ns of the trajectories.

**MtD simulations**. The reference path was built upon the different conformations of the nucleotides U285 to A290 in the pre-hydrolytic and toggled states (PDB id: 4FAQ and 4FAX, respectively)[16]. The two structures were used to generate 30 interpolated intermediates through the MolMov morphing server[70]. Each intermediate was subjected to energy minimization, and 16 snapshots were chosen to build the path. Each node of the path (i.e., intermediate structure) is equally spaced with a distance of ~0.32 Å. According to Branduardi et al.[71], we defined two-path collective variables: (1) S, which defines the progress along the reference path; (2) Z, which measures the distance from the reference path. To sample the free energy landscape, we used adaptative-width MtD as implemented in Plumed[35,72], in which the width of the gaussian was determined by the fluctuation of S and Z over a time interval of 1 ps. A lower-bound limit for the width of the gaussian was set to 0.03 in the appropriate unit for each coordinate. The height of the gaussian was set to 0.3 kJ/mol with an additional frequency of 1 ps. By considering the distance between the nodes of the path, we set a $\lambda = 23.66$ Å$^{-2}$. We collected: (1) 350 ns, for the transition cleaved state to toggled state (referred to as $c \rightarrow T$); (2) 200 ns for the transition cleaved-H⁺ state to the toggled state (referred to as $cH^+ \rightarrow T$).

**Hybrid quantum mechanical/molecular mechanical (QM/MM) simulations**. QM/MM simulations were performed on the structure of the cleaved-H⁺ state with CP2K molecular dynamics engine[73] to explore the stability of the protonated form of N3$^{C358}$. The AMBER force field was used for the MM subsystem, whereas density functional theory (DFT) was used to describe the QM atoms. The BLYP functional[74,75] supplemented by a dispersion correction was employed[76]. The Quickstep algorithm was used to solve the electronic structure problem[77], employing a double zeta plus polarization basis set[78] to represent the valence orbitals and plane waves for the electron density (320 Ry cutoff). Goedecker–Teter–Hutter type pseudopotentials were used for valence–core interactions[79]. Wavefunction optimization was achieved through an orbital transformation method[80] using a threshold of $5 \cdot 10^{-7}$ on the electronic gradient as a convergence criterion. The QM/MM coupling follows the protocol proposed by Laino et al.[81]. Simulations were performed in the NVT ensemble (300 K), employing a velocity rescaling thermostat[68]. After about 4.3 ps, a second water molecule was included in the QM region, and the simulation was restarted to collect 15 ps of simulation time. N3$^{C358}$ remained stably protonated throughout the entire simulation.

**Reporting summary**. Further information on research design is available in the Nature Research Reporting Summary linked to this article.

## Data availability
Data supporting all other findings of this manuscript, including MD simulation trajectories, are available from the corresponding authors upon request. Coordinates and structure factors have been deposited in the Protein Data Bank under accession codes PDB 6T3K, PDB 6T3R, PDB 6T3N, and PDB 6T3S. The source data underlying Fig. 1b, c and Supplementary Table 2 are provided as a Source Data file. Source data are provided with this paper.

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

## Acknowledgements

This work is based upon research conducted at the Northeastern Collaborative Access Team beamlines, which are funded by the National Institute of General Medical Sciences from the National Institutes of Health (P30 GM124165). The Eiger 16M detector on 24-ID-E is funded by a NIH-ORIP HEI grant (S10OD021527). This research used resources of the Advanced Photon Source, a U.S. Department of Energy (DOE) Office of Science User Facility operated for the DOE Office of Science by Argonne National Laboratory under Contract No. DE-AC02-06CH11357. We would also like to thank Dr Laura Murray for help in cloning and crystallizing the G and U mutants, Gabriele Drews for technical assistance, and Dr. Olga Fedorova for critical reading of the manuscript. We also thank all members of the Marcia, Pyle, and De Vivo labs for helpful discussion. Work in the Marcia lab is partly funded by the Agence Nationale de la Recherche (ANR-15-CE11-0003-01), by the Agence Nationale de Recherche sur le Sida et les hépatites virales (ANRS) (ECTZ18552), by ITMO Cancer (18CN047-00), and by the Fondation ARC pour la recherche sur le cancer (PJA20191209284). The Marcia lab uses the platforms of the Grenoble Instruct Center (ISBG: UMS 3518 CNRS-CEA-UJF-EMBL) with support from FRISBI (ANR-10-INSB-05-02) and GRAL (ANR-10-LABX-49-01) within the Grenoble Partnership for Structural Biology (PSB). MDV thanks the Italian Association for Cancer Research (AIRC) for financial support (IG 23679). AMP is a Howard Hughes Medical Institute investigator.

## Author contributions

MM, AMP, and MDV have conceived and designed the work; SS performed initial $pK_A$ calculations; JM, IC, MM, VG, and PV have acquired the data; all authors have interpreted the data and drafted the manuscript. MM and MDV contributed equally.

## Competing interests

The authors declare no competing interests.
