## [Peer Review File · Nature Communications]

Reviewers' comments:

Reviewer #1 (Remarks to the Author):

This is a nice paper based on a combination of X-Ray crystallography and MD simulation using state of the art protocols. The work is done by two excellent groups, the topic is of interest, and the analysis is quite complete. Unfortunately, I am not convinced that the proposed mechanism is really supported by the results presented here.

Experimental results in Fig 1 shows a significant decrease in the G-mutant, but there is not such an impressive decay in the U-mutant, where no acceptor site exist. The ratio k_1 WT/U-mutant is around 3, and the same ratio for the k_2 is around 4 (within the error bar). Thus, there is not difference in the speed of the first and second step due to the U-mutation. This does not agree with the need of a C358 acting as a base.

The distance between proton donor and proton acceptor in the suggested mechanism is very large (close to 10 Å; i.e. 3 water molecules) and no evidence from calculation or experiment is provided that this long proton channeling is possible.

Figure 2 and 3 show very little differences in the mutants in terms of the RNA, backbone, but it is not clear the ion environment, whether or not ions present in the figure have always a clearly associated electron density (for example Figure 3 shows spheres labelled as K1 and K2, when Na⁺ rather than K⁺ was in the crystallization buffer, and there are not electron density plots to convince the reader that the ions are really there. So, the crucial role of monovalent ions suggested by simulation is not so clear based on the X-ray. The difference in WT between Na⁺ and K⁺ at the WT triad is quite surprising and not explained. It might be a Na⁺ effect, but both G- and U-mutant recover the triad conformation.

The pKa of C358 is normal in the pre-hydrolytic state and around neutral at the toggled state (this is not unexpected). However, the pKa shifts in C377 are too extreme (nearly 10 units) and no explanation of this dramatic shift is presented in the paper. These changes are not easy to believe, especially as they were suggested by Poisson Boltzman calculations which are very susceptible to geometrical errors and to uncertainties in the definition of the grid or the dielectrics.

In summary, I enjoyed the elegant mechanism suggested and it might be true, but I am afraid it is not supported by the results presented in the paper.

Reviewer #2 (Remarks to the Author):

In this manuscript entitled “Visualizing group II intron dynamics between the first and second steps of splicing”, the authors employed biochemistry, crystallography and MD simulation methodologies to address the mechanism coordinating the two steps of group II intron splicing by hydrolysis. The experiments were well designed and executed, the arguments are clear and the figures and the movie are illustrative. However, a few aspects of the manuscript should be addressed before publication.

General Comments:

1. It would be good to introduce M1, M2, K1 and K2 in the Introduction to help the reader familiarize those key ions.
2. Page 3 line 10: references should include relevant work reported by the Lambowitz group.
3. A sentence of caveat should be given regarding possible existence of other factors in addition to the ability to protonate that may result from replacing C358 with G or U.
4. Page 7 line 25: “establish” should be changed to “suggest”, as the connection between C358 protonation and active site toggling is inferred, not proven.
5. Figure 1 data show impaired first step of splicing as well as reduced second-step splicing for the mutants, especially for the U-mutant. This must be discussed. In addition, it would be beneficial to discuss the different splicing efficiencies of the two mutants.
6. It would be helpful to readers not familiar with the O.i. intron to include a secondary structure map of the intron highlighting key nucleotides (C358, C377, etc.)
7. The ionic strength of the MD simulation environment should be specified, in addition to refs cited (Joung et al. and Li et al.). It should also be made clearer whether the same conditions were used in the experimental work of the paper.
8. It would be interesting to discuss the implications of the results to lariat-forming group II introns, as those are mechanistically more relevant to spliceosomes than splicing by hydrolysis. In addition, it would be good to distinguish hydrolysis and generation of linear intron from lariat-formation.

Minor points:

1. Page 6 line 22: RMSD should be defined as it first appears.
2. Page 11 line 9: “associated to” should be changed to “associated with”.
3. Page 15 line 20: “such key” should be changed to “such a key”.
4. In the legends of Figures 2 and 3, “J23” should be changed to “J2/3”.
5. “Retrotransposition” in Figure 7 is more appropriate as “retromobility”.

Reviewed by: Marlene Belfort and Eren Dong

Reviewer #3 (Remarks to the Author):

Manigrasso et al. describe a combined enzymatic, crystal structural and computational study of self-splicing group II introns, with a particular focus on the transition from the first to the second step of splicing. The authors show that substitution of a hitherto unexplored active site residue, C358, with G or U predominantly leads to defects in the second step of splicing, consistent with C358 being transiently protonated between the two steps. They determined crystal structures of the two mutants (in which also other residues were replaced to maintain the possibility for proper active site formation) in the presence of potassium/magnesium or sodium/magnesium. The structures revealed that the mutants did not suffer detrimental structural changes in their active sites that could explain differences in their activities. They also showed that the mutants were inhibited with respect to adopt a “toggled” conformation seen in the presence of sodium with the wt intron. The authors then conducted extensive molecular dynamics simulations and free energy calculations, which revealed a correlation between C358 protonation and toggling. In particular, they observed a series of local conformational changes in the protonated state (and more slowly in the non-protonated state) of the wt intron that lead to release of one active site potassium ion. The mutants did not show similar rearrangements in MD simulations. Together, the results delineate specific active site rearrangements in preparation of step 2 and how these rearrangements depend on the ability of C358 to be protonated.

The manuscript reports interesting new findings that are presented in a clear and systematic manner and that should thus be accessible to a large audience. Apart from the molecular mechanisms of group II introns, the results also have important implications for nuclear pre-mRNA splicing by the spliceosome. The biochemical and crystal structural analyses seem to have been expertly conducted. This reviewer cannot comment on the validity of the MD simulations/energy calculations, but the results of these complex analyses have certainly been reported clearly and effectively. The manuscript is very well written and the Figures are effectively designed.

This reviewer has no specific comments.

Visualizing group II intron dynamics between the first and second steps of splicing

Jacopo Manigrasso^{1,#}, Isabel Chillón^{2,#}, Vito Genna³, Pietro Vidossich¹, Srinivas Somarowthu⁴, Anna Marie Pyle^{3,6,7}, Marco De Vivo^{1,*}, Marco Marcia^{2,*}

*To whom correspondence should be addressed. [E-mail: marco.devivo@iit.it](mailto:marco.devivo@iit.it); mmarcia@embl.fr

REVIEWER #1:

This is a nice paper based on a combination of X-Ray crystallography and MD simulation using state of the art protocols. The work is done by two excellent groups, the topic is of interest, and the analysis is quite complete. Unfortunately, I am not convinced that the proposed mechanism is really supported by the results presented here.

Experimental results in Fig 1 shows a significant decrease in the G-mutant, but there is not such an impressive decay in the U-mutant, where no acceptor site exist. The ratio k_1 WT/U-mutant is around 3, and the same ratio for the k_2 is around 4 (within the error bar). Thus, there is not difference in the speed of the first and second step due to the U-mutation. This does not agree with the need of a C358 acting as a base.

We would like to thank the reviewer for the general appreciation of our research and of the completeness of our analysis.

Considering the reviewer's concerns about the kinetics of the mutants, we would like to make the following considerations.

First, while we agree with the reviewer that the differences between the first and second splicing step rates are less pronounced for the U-mutant than for the G-mutant, we would like to point out that both mutants show accumulation of linear intron/3'-exon intermediate compared to wild type (Fig 1C, middle panel). This experimentally-observed accumulation of intermediate is the parameter commonly used in the group II intron field to unambiguously show intron stalling after the first step of splicing. This is a critical point that should be fully appreciated and that we emphasize now in the text in response to this criticism. Crucially, to reinforce our interpretations of the kinetics results, we have now performed new splicing assays on the newly-cloned "A-mutant", a mutant that replaces C358 with adenosine, and can thus be protonated at position 358. As reported in the newly-modified Fig 1, the A-mutant splices at rates comparable to wild type. Therefore, our new data prove that the splicing defects are exclusively observed in the G- and U-mutants, the only two mutants that cannot be protonated. We have modified our main text in the Results section (page 6, lines 8-16) to explain and underline this key phenomenon more clearly.

As further support to our mechanistic hypothesis and interpretation of our kinetic data, we would also like to point out that the splicing defects of our mutants (~12-fold for the G-mutant; ~7-fold for the U-mutant) are not only very significant and outside our experimental error bars with respect to the A-mutant and to wild type, but they are also in perfect agreement with the splicing defects of analogous mutants described in previous literature. For instance, in pioneering work for group II intron splicing, Peebles and colleagues had quantified the splicing defects of constructs of the *ai5γ* intron from *S. cerevisiae* mutated at the catalytic triad position analogous to position 358 of our *O. theyensis* intron (Peebles *et al.*, 1995). In that work, the G-mutant showed 15-fold and the U-mutant 7-fold defects compared to wild type (which carries an A). The C-mutant instead spliced at wild type rates. In our approach, we have modified the active site more gently and accurately than Peebles *et al.* introducing triple mutations of the triad to restore the Watson Crick base pair of nucleotide 358 and its J2/3 junction

pair (the existence of a triple helix involving J2/3 was not yet known when the Peebles paper was published). It is thus unreasonable to expect that our mutant would show stronger splicing defects than the Peebles mutants. We now report the comparison with the Peebles work in the Results section (page 6, lines 16-18) and in the Discussion section (page 15, lines 19-20).

This said, we agree with the reviewer that our G- and U-mutants are not completely inactive, but only slower in splicing than wild type. Interestingly, this result is in line with the rest of our experimental data. Our crystal structures show that the G- and U-mutants do undergo the first step of splicing (Fig 2) and our MD simulations show that toggling can occur in wild type even in the absence of protonation (Fig 5, S5, and S6C; main text, page 9 lines 26-28). In light of these observations, it is more accurate to state in our revised text that our enzymatic, crystallographic and computational data all point to the fact that 358 protonation is not an absolutely necessary event for catalysis, but an important event because it favors splicing. We have thus specifically modified the text in the abstract (page 2, line 11), the highlights (page 2, lines 20-22), the results section (page 8, line 4), and the discussion section (page 13, lines 17/20-28; page 14, lines 1-6; page 16, line 15).

The distance between proton donor and proton acceptor in the suggested mechanism is very large (close to 10 Å; i.e. 3 water molecules) and no evidence from calculation or experiment is provided that this long proton channeling is possible.

In response to this concern of the reviewer, we would like to make the following considerations.

First, while we agree with the reviewer that based on our proposed mechanism proton donor (the reaction nucleophile) and acceptor (the N3 atom of C358) are 9.8 Å apart, we would like to mention that this is not an uncommon distance for proton transfer pathways. Most importantly, we did not propose proton transfer through bulk solvent as a speculative assumption but based on a well-established mechanism previously proven for the same group II intron by Casalino *et al.* (Casalino *et al.*, 2016). In that work, the authors have performed QM/MM calculations to describe proton channeling post-5'-splicing hydrolysis and describe proton transfer pathways “*involving up to five water molecules*”, which corresponds to a migration distance of ~15 Å. The energy barriers involved in those pathways are compatible with experimental catalytic rates. To complement the QM/MM calculations of Casalino *et al.*, we have now performed new QM/MM calculations ourselves on the protonated form of C358 and we have proven that protonated C358 is very stable, maintaining its proton stably bound to atom N3 for over 15 ps of simulation (new Fig S2B-C). We have now added these considerations more explicitly in the text (page 5, lines 15-22). Our new QM/MM calculations have been performed by Dr. Pietro Vidossich, who has thus now been added to the list of contributing authors.

Second, long proton transfer pathways and related energetics are not uncommon in enzymes. For instance, in ribonuclease H (RNase H), an enzyme with important structural and mechanistic analogies to the group II intron (Genna *et al.*, 2018), one proton is transferred from the nucleophilic oxygen to the leaving phosphate via three or four water molecules over a distance of 6-7 Å during catalysis (De Vivo *et al.*, 2008). Besides RNase H, long-distance and water-mediated proton transfers are well characterized in the enzymatic conversion of chromopyrrolic acid to an antitumor derivative by cytochrome P450, where computations and experiments have shown the importance of proton transfers through waters (Wang *et al.*, 2009). Analogously, many membrane proteins, including respiratory and photosynthetic complexes, use long water chains to conduct protons. For instance, in the water-oxidizing photosystem II (PSII) a long-distance proton transfer occurs along an extended water chain for over 13.5 Å (Saito *et al.*, 2015; Takaoka *et al.*, 2016).

Finally, we would like to clarify that identifying with precision the exact proton transfer pathway that connects the reaction nucleophile to C358 does not impact on the main findings of our work. Indeed, several other possible mechanisms for proton migration may exist, e.g. a transfer via a different network of water molecules or diffusion of a proton in the bulk solvent, and all the possible mechanisms are compatible with the splicing mechanism we propose. As a matter of fact, independent of how the proton is exactly transferred to C358, our mechanism remains valid and corroborated by the splicing kinetics and by the large amount of equilibrium and non-equilibrium force-field-based simulations of the group II intron that we have performed in different protonation states and at various stages of catalysis (Fig 46 and S3-S9). To clarify this key point and fully account for alternative, mechanistically equivalent possibilities of proton migration, we have modified the main text (page 13, lines 9-14) and Fig S2 legend.

Figure 2 and 3 show very little differences in the mutants in terms of the RNA, backbone, but it is not clear the ion environment, whether or not ions present in the figure have always a clearly associated electron density (for example Figure 3 shows spheres labelled as K1 and K2, when Na⁺ rather than K⁺ was in the crystallization buffer, and there are not electron density plots to convince the reader that the ions are really there. So, the crucial role of monovalent ions suggested by simulation is not so clear based on the X-ray. The difference in WT between Na⁺ and K⁺ at the WT triad is quite surprising and not explained. It might be a Na⁺ effect, but both G- and U-mutant recover the triad conformation.

In response to this comment of the reviewer, we would like to clarify that all ions for which there is no electron density are not present in the structure. In our original version of the manuscript, we had indicated those ions in semi-transparent representation, as explained in the figure legend, to help the reader localize their position in the image. However, since we now realize that such representation is a source of confusion, we have modified Fig 3 and we have removed all ions which are absent in the actual crystallographic structures.

Concerning the structural differences in wild type between the sodium and potassium conditions, we would like to remark that an extensive discussion of such features and their catalytic implications is included in our previous work (Marcia and Pyle, 2012), and that our current manuscript is directly based on it. We have expanded our text (page 7, lines 23-24) to reinforce the description of the wild type behavior in sodium vs potassium and to indicate the reference to our previous work more explicitly.

Finally, as remarked by the reviewer, both the G- and the U-mutant adopt the triple helix conformation in sodium and in potassium, differently from wild type. This observation is exactly what suggested to us that the G- and U-mutants have difficulty in adopting the toggled conformation, which may explain their tendency to stall after the first step of splicing and their difficulties in progressing towards the second step. These structural data imply a connection between position 358 and active site toggling, which is an important step to properly rearrange the intron active site between the two steps of splicing. We have now rephrased the text (page 8, line 2) to clarify this concept.

The pK_a of C358 is normal in the pre-hydrolytic state and around neutral at the toggled state (this is not unexpected). However, the pK_a shifts in C377 are too extreme (nearly 10 units) and no explanation of this dramatic shift is presented in the paper. These changes are not easy to believe, especially as they were suggested by Poisson Boltzman calculations which are very susceptible to geometrical errors and to uncertainties in the definition of the grid or the dielectrics. In summary, I enjoyed the elegant mechanism suggested and it might be true, but I am afraid it is not supported by the results presented in the paper.

We fully agree with the reviewer that Poisson-Boltzmann calculations of pK_a shifts are susceptible to geometrical errors and uncertainties in the computational parameters. Indeed, we used those calculations only as a qualitative and merely indicative initial reference to assess the consequences of the pronounced structural changes that we observed in our different crystallographic states. We have now expressed these considerations explicitly in the text to warn the reader that our electrostatic calculations are only qualitative (page 5, lines 10-11). With these warnings, it should be clear to the reader that no conclusions could be derived from electrostatic calculations only. However, we would like to remark that our conclusions and mechanistic model are not based on the electrostatic calculations, but on our experimental results from splicing kinetics, crystal structures and computational simulations, which we obtained in a totally independent manner from the Poisson-Boltzmann calculations and which are in perfect reciprocal agreement.

We have also rephrased the text by removing any consideration related to C377, which is not the subject of investigation of the present manuscript. C377 was instead extensively probed in our previous work, where we had demonstrated its involvement in catalysis and the detailed conformational changes that it undergoes during the transition from the first to the second step of splicing (Marcia and Pyle, 2012).

REVIEWER #2:

This In this manuscript entitled “Visualizing group II intron dynamics between the first and second steps of splicing”, the authors employed biochemistry, crystallography and MD simulation methodologies to address the mechanism coordinating the two steps of group II intron splicing by hydrolysis. The experiments were well designed and executed, the arguments are clear and the figures and the movie are illustrative. However, a few aspects of the manuscript should be addressed before publication.

General Comments:

1. It would be good to introduce M1, M2, K1 and K2 in the Introduction to help the reader familiarize those key ions.

We would like to thank the reviewer for the appreciation of our manuscript.

To address general comment 1, we have expanded the Introduction section with specific comments on the role and location of M1-M2-K1-K2 (page 3, lines 20-24 and page 4, line 5)

2. Page 3 line 10: references should include relevant work reported by the Lambowitz group.

To address the reviewer’s comment, we have now included the following references from the Lambowitz group on page 3, line 10:

- Cui, X., Matsuura, M., Wang, Q., Ma, H. & Lambowitz, A.M. A group II intron-encoded maturase functions preferentially in cis and requires both the reverse transcriptase and X domains to promote RNA splicing. *J Mol Biol* 340, 211-31 (2004).
- Matsuura, M., Noah, J.W. & Lambowitz, A.M. Mechanism of maturase-promoted group II intron splicing. *EMBO J* 20, 7259-70 (2001).

3. A sentence of caveat should be given regarding possible existence of other factors in addition to the ability to protonate that may result from replacing C358 with G or U.

We thank the reviewer for this suggestion and to address their comments, we have added a sentence on page 13, lines 25-28, stating that steric or electrostatic perturbations of the active site may partially account for the measured splicing defects, besides the inability of the G- and U-mutants to be protonated at position 358.

4. Page 7 line 25: “establish” should be changed to “suggest”, as the connection between C358 protonation and active site toggling is inferred, not proven.

We have edited the text as recommended by the reviewers.

5. Figure 1 data show impaired first step of splicing as well as reduced second-step splicing for the mutants, especially for the U-mutant. This must be discussed. In addition, it would be beneficial to discuss the different splicing efficiencies of the two mutants.

To address the reviewer’s comment, we have edited the text in the Results section on page 6, lines 822, where we now report the first step of splicing defects of both the G- and the U-mutants, along with the second step of splicing defects. We have now also measured splicing kinetics of the newly-cloned A-mutant, which splices at rates comparable to wild type. This new result is important because it shows that splicing defects are limited to non-protonatable mutants (G- and U-mutants), but do not occur in protonatable mutants (A-mutant). Finally, we specifically note that the splicing defects of the G- and U-mutants reported in our work are perfectly in line with previous literature in the field (Peebles *et al.*, 1995) (see also our response to the first comment of reviewer 1, above).

Additionally, we have edited the text in the Discussion section on page 13, lines 20-25. Here, we explain that both the kinetic assays and the MD simulations of the cleaved post-hydrolytic state show a very consistent phenomenon. The splicing reaction can occur in the absence of protonation on 358, but protonation favors it by accelerating it significantly in wild type introns. This difference in kinetics may constitute a sufficiently strong phenotypic advantage for the intron to have preserved protonatable

residues at position 358 throughout evolution.

6. It would be helpful to readers not familiar with the O.i. intron to include a secondary structure map of the intron highlighting key nucleotides (C358, C377, etc.).

To address the reviewer's comment, we have included the secondary structure map of the intron in Fig 2, panel E.

7. The ionic strength of the MD simulation environment should be specified, in addition to refs cited (Joung et al. and Li et al.). It should also be made clearer whether the same conditions were used in the experimental work of the paper.

In the MD simulations, we have used ionic concentrations of 100 mM for magnesium ions and 150 mM for potassium ions. These values, which are in line with the reported crystallization conditions, are now reported in the Material and Methods section (page 20, lines 1-3).

8. It would be interesting to discuss the implications of the results to lariat-forming group II introns, as those are mechanistically more relevant to spliceosomes than splicing by hydrolysis. In addition, it would be good to distinguish hydrolysis and generation of linear intron from lariat-formation.

To address the reviewer's comment, we have included a more specific reference to the hydrolytic and transesterification pathways in the Introduction (page 3, lines 20-21) along with a reference that describes those mechanisms in detail (Pyle, 2010). Additionally, in the Discussion section (page 15, lines 15-20), we have discussed the applicability of our protonation and toggling dependent mechanism to introns that splice via hydrolysis and introns that splice via transesterification. Briefly, because the two pathways follow the same reaction chemistry, and because biochemical and structural data suggest that they involve the same active site residues positioning the nucleophile in the exact same structural position compared to nucleotide 358, we find it very plausible that our mechanism may apply to the transesterification splicing pathway. Indeed, analogous mutants cause similar splicing defects in the lariat-forming ai5 γ intron (see also our response to General point 5 of reviewer 2, above). Finally, the analogies that we report between our mechanism – derived for a hydrolytic intron – and the spliceosome – which follows the transesterification pathway – additionally reinforces our conclusions.

Minor points:

1. Page 6 line 22: RMSD should be defined as it first appears.

To address the reviewer's comment, we have defined RMSD as root mean square deviation in its first instance in the text (currently on page 7, line 3).

2. Page 11 line 9: "associated to" should be changed to "associated with".

To address the reviewer's comment, we have changed "associated to" to "associated with" in the text (currently on page 11, line 15).

3. Page 15 line 20: "such key" should be changed to "such a key".

To address the reviewer's comment, we have changed "such key" to "such a key" in the text (currently on page 16, line 14).

4. In the legends of Figures 2 and 3, "J23" should be changed to "J2/3". To address the reviewer's comment, we have changed "J23" to "J2/3" in the legends of Fig 2 and 3.

5. "Retrotransposition" in Figure 7 is more appropriate as "retromobility". To address the reviewer's comment, we have changed "retrotransposition" to "retromobility" in Fig 7.

REVIEWER #3:

Manigrasso et al. describe a combined enzymatic, crystal structural and computational study of self-splicing group II introns, with a particular focus on the transition from the first to the second step of splicing. The authors show that substitution of a hitherto unexplored active site residue, C358, with G or U predominantly leads to defects in the second step of splicing, consistent with C358 being transiently protonated between the two steps. They determined crystal structures of the two mutants (in which also other residues were replaced to maintain the possibility for proper active site formation) in the presence of potassium/magnesium or sodium/magnesium. The structures revealed that the mutants did not suffer detrimental structural changes in their active sites that could explain differences in their activities. They also showed that the mutants were inhibited with respect to adopt a “toggled” conformation seen in the presence of sodium with the wt intron. The authors then conducted extensive molecular dynamics simulations and free energy calculations, which revealed a correlation between C358 protonation and toggling. In particular, they observed a series of local conformational changes in the protonated state (and more slowly in the non-protonated state) of the wt intron that lead to release of one active site potassium ion. The mutants did not show similar rearrangements in MD simulations. Together, the results delineate specific active site rearrangements in preparation of step 2 and how these rearrangements depend on the ability of C358 to be protonated.

The manuscript reports interesting new findings that are presented in a clear and systematic manner and that should thus be accessible to a large audience. Apart from the molecular mechanisms of group II introns, the results also have important implications for nuclear pre-mRNA splicing by the spliceosome. The biochemical and crystal structural analyses seem to have been expertly conducted. This reviewer cannot comment on the validity of the MD simulations/energy calculations, but the results of these complex analyses have certainly been reported clearly and effectively. The manuscript is very well written and the Figures are effectively designed.

This reviewer has no specific comments.

We would like to thank the reviewer for the great appreciation of our manuscript, which is very encouraging.

REFERENCES:

1. Casalino L., Palermo G., Rothlisberger U., Magistrato A. (2016) Who activates the nucleophile in ribozyme catalysis? An answer from the splicing mechanism of group II introns. *J Am Chem Soc* 138:10374-10377.
2. De Vivo M., Dal Peraro M., Klein M. L. (2008) Phosphodiester cleavage in ribonuclease H occurs via an associative two-metal-aided catalytic mechanism. *J Am Chem Soc* 130:10955-10962.
3. Genna V., Colombo M., De Vivo M., Marcia M. (2018) Second-Shell Basic Residues Expand the Two-Metal-Ion Architecture of DNA and RNA Processing Enzymes. *Structure* 26:40-50 e42.
4. Marcia M., Pyle A. M. (2012) Visualizing group II intron catalysis through the stages of splicing. *Cell* 151:497-507.
5. Peebles C. L., Zhang M., Perlman P. S., Franzen J. S. (1995) Catalytically critical nucleotide in domain 5 of a group II intron. *Proc Natl Acad Sci U S A* 92:4422-4426.
6. Pyle A. M. (2010) The tertiary structure of group II introns: implications for biological function and evolution. *Crit Rev Biochem Mol Biol* 45:215-232.
7. Saito K., Rutherford A. W., Ishikita H. (2015) Energetics of proton release on the first oxidation step in the water-oxidizing enzyme. *Nat Commun* 6:8488.
8. Takaoka T., Sakashita N., Saito K., Ishikita H. (2016) pK(a) of a Proton-Conducting Water Chain in Photosystem II. *J Phys Chem Lett* 7:1925-1932.
9. Wang Y., Chen H., Makino M., Shiro Y., Nagano S., Asamizu S., Onaka H., Shaik S. (2009) Theoretical and experimental studies of the conversion of chromopyrolic acid to an antitumor derivative by cytochrome P450 StaP: the catalytic role of water molecules. *J Am Chem Soc* 131:6748-6762.

Reviewers' comments:

Reviewer #2 (Remarks to the Author):

OK with the changes that addressed my comments.

Reviewer #4 (Remarks to the Author):

In this paper, Marcia and coworkers perform a detailed biochemical, structural and simulation analysis of group II intron dynamics between the first and second steps of splicing. This paper addresses a biochemically important problem, and the work is generally well executed. As the work has already been extensively reviewed, I don't have too many additional comments, except that something that is missing from the paper is a mechanistic figure, it is rather hard to follow the discussion without a proper mechanistic figure. It can be in the SI if the authors prefer and there are space limitations, but it is important that it is included so readers can follow the mechanistic discussion in the main text. I note that on pg. 3, the authors are in fact referring to such a figure (Figure 1), which has now been replaced by a figure showing kinetic data. Figure 7 fulfils this request to some extent, but something like a clear ChemDraw sketch would be very useful, especially as the reader doesn't get to Figure S7 until very late in the paper.

I do also have some comments with regard to specific changes in response to Reviewer 1:

- The introduction of the new A-mutant is important. I agree with Reviewer 1's concerns about the similarity in the kinetics of the WT to each of the G/U mutants is a cause for concern, and I think showing that the A-mutant (which, like the wild-type is able to be protonated) has similar splicing rates to the wild-type is important, in distinguishing the G/U mutants from the wild-type. However, I have a new concern: ~12-fold and ~7-fold splicing defects compared to wild-type and each of the G- and U-mutants, respectively, are much smaller effects than I would have expected for removal of a general-base, if the general-base is important (which I would have expected to abolish activity by several orders of magnitude). The authors say themselves in the rebuttal that this protonation step is not absolutely necessary but an important step, but I think even the word 'important' is quite strong for something that has only ~10-fold impact on the rate. I would clarify this further in the manuscript. In addition, can the authors discuss more why even the A-mutant (which can be protonated) leads nevertheless to a ~2-fold loss in activity compared to WT?
- With regard to Reviewer 1's concern about the water chain, I am inclined to agree with the authors: as long as there is a clear proton-transfer chain these proton transfer events should be fast, so I am not as concerned about the distance. However, one would expect proton transfer through a long chain to still be less efficient than direct proton transfer, and that might be commented on.
- The issue with regard to ion density has been addressed satisfactorily.

- I fully agree with the reviewer about a 10 pKa unit difference being far too extreme. This is not surprising however, as empirical estimates of pKas can grossly overestimate pKa shifts, and pKa shifts in general are notoriously difficult to correctly calculate using any approach. I think the authors need to be even more explicit in stating that the values in Table S1 are clearly overestimates, but can provide a qualitative approximation of the direction of pKa shifts, even if they are not so quantitatively reliable. In addition, considering how extreme the overestimate of the downward pKa shift of C377 has been, the upward pKa shift of C358 is likely also over-exaggerated, so all mention of 'drastic shifts' (including in the section header) need to be removed, and replaced with just the comment that the calculations indicate a likely shift in pKa.

Lynn Kamerlin

RESPONSE TO REVIEWERS COMMENTS FOR:

Visualizing group II intron dynamics between the first and second steps of splicing

Jacopo Manigrasso^{1,#}, Isabel Chillón^{2,#}, Vito Genna³, Pietro Vidossich¹, Srinivas Somarowthu⁴, Anna Marie Pyle^{5,6,7}, Marco De Vivo^{1,*}, Marco Marcia^{2,*}

*To whom correspondence should be addressed. E-mail: marco.devivo@iit.it; mmarcia@embl.fr

We would like to thank the editor and reviewers #2 and #4 for their further comments on our manuscript. In this detailed point-by-point response, we describe how we have addressed the new comments of reviewer #4. We also enclose a revised version of the manuscript and figures, in which new revisions are marked in red.

REVIEWER #2:

OK with the changes that addressed my comments.

Thank you.

REVIEWER #4:

In this paper, Marcia and coworkers perform a detailed biochemical, structural and simulation analysis of group II intron dynamics between the first and second steps of splicing. This paper addresses a biochemically important problem, and the work is generally well executed. As the work has already been extensively reviewed, I don't have too many additional comments, except that something that is missing from the paper is a mechanistic figure, it is rather hard to follow the discussion without a proper mechanistic figure. It can be in the SI if the authors prefer and there are space limitations, but it is important that it is included so readers can follow the mechanistic discussion in the main text. I note that on pg. 3, the authors are in fact referring to such a figure (Figure 1), which has now been replaced by a figure showing kinetic data. Figure 7 fulfils this request to some extent, but something like a clear ChemDraw sketch would be very useful, especially as the reader doesn't get to Figure S7 until very late in the paper.

We would like to thank the reviewer for the appreciation of our work and for the suggestion to add a ChemDraw sketch of the splicing reaction. To address this comment, we have modified Figure 1A, where we now include sketches of precursor (5e-I-3e), intermediate (I-3e) and spliced intron/ligated exons (I + 5e-3e), along with the schematics of the nucleophilic attacks that occur during splicing. This new sketch in Figure 1A should help the reader follow the Introduction section of our manuscript. We have additionally added a sketch of the protonated and toggled state described in our work to Figure 7 (new panel B). This panel should help the reader follow our Discussion section. By comparison to the new Figure 1A, the new Figure 7B should also facilitate the identification of the protonatable residue C358 and the toggling residue G288 with respect to the rest of the active site. We now report the kinetic rate constants of all constructs in the new Table S2. We have renumbered Figures and Tables throughout the manuscript to account for these changes.

I do also have some comments with regard to specific changes in response to Reviewer 1:

- The introduction of the new A-mutant is important. I agree with Reviewer 1's concerns about the similarity in the kinetics of the WT to each of the G/U mutants is a cause for concern, and I think showing that the A-mutant (which, like the wild-type is able to be protonated) has similar splicing rates to the wild-type is important, in distinguishing the G/U mutants from the wild-type. However, I have a new concern: ~12-fold and ~7-fold splicing defects compared to wild-type and each of the G- and U-mutants, respectively, are much smaller effects than I would have expected for removal of a general-base, if the general-base is important (which I would have expected to abolish activity by several orders of magnitude). The authors say themselves in the rebuttal that this protonation step is not absolutely necessary but an important step, but I think even the word 'important' is quite

strong for something that has only ~10-fold impact on the rate. I would clarify this further in the manuscript. In addition, can the authors discuss more why even the A-mutant (which can be protonated) leads nevertheless to a ~2-fold loss in activity compared to WT?

In response to this concern of the reviewer, we have rephrased our text and removed the word “important” in the Abstract (page 2, line 11-12), the Results section (page 8, lines 6-7), and the Discussion section (page 14, line 17). We have also rephrased the Discussion section (page 14, lines 26-28; page 15, lines 1-5) to describe that steric or electrostatic perturbations may explain why the A-mutant may be ~2-fold less active than WT. We would like to stress the fact that the A-mutant crucially does not accumulate linear intron/3'-exon intermediate (Figure 1C, middle panel), which is the parameter commonly used in the group II intron field to unambiguously show intron stalling after the first step of splicing. The A-mutant could thus serve as a qualitative reference to estimate the impact of steric/electrostatic perturbations introduced by our mutations on splicing rates. This impact is limited (~2-fold defects, no accumulation of I-3e) and thus reinforce our conclusions that the defects of the G- and U-mutants are predominantly due to their inability of being protonated at position 358.

- With regard to Reviewer 1's concern about the water chain, I am inclined to agree with the authors: as long as there is a clear proton-transfer chain these proton transfer events should be fast, so I am not as concerned about the distance. However, one would expect proton transfer through a long chain to still be less efficient than direct proton transfer, and that might be commented on.

In response to this comment of the reviewer, we have rephrased our text in the Results section (page 5, lines 18-19), to explain that direct proton transfer would be faster than long-distance transfer through chains of water molecules.

- The issue with regard to ion density has been addressed satisfactorily.

Thank you.

- I fully agree with the reviewer about a 10 pKa unit difference being far too extreme. This is not surprising however, as empirical estimates of pKas can grossly overestimate pKa shifts, and pKa shifts in general are notoriously difficult to correctly calculate using any approach. I think the authors need to be even more explicit in stating that the values in Table S1 are clearly overestimates, but can provide a qualitative approximation of the direction of pKa shifts, even if they are not so quantitatively reliable. In addition, considering how extreme the overestimate of the downward pKa shift of C377 has been, the upward pKa shift of C358 is likely also over-exaggerated, so all mention of 'drastic shifts' (including in the section header) need to be removed, and replaced with just the comment that the calculations indicate a likely shift in pKa.

In response to this comment of the reviewer, we have rephrased our text in the Results section (page 5, lines 2-3/5/8) and in the legend of Table S1 to emphasize that the pK_A values are only qualitative estimates.

REVIEWERS' COMMENTS:

Reviewer #4 (Remarks to the Author):

The authors have addressed all my additional comments on the manuscript, and I am satisfied with their revisions.

Lynn Kamerlin

RESPONSE TO REVIEWERS COMMENTS FOR:

Visualizing group II intron dynamics between the first and second steps of splicing

Jacopo Manigrasso^{1,#}, Isabel Chillón^{2,#}, Vito Genna³, Pietro Vidossich¹, Srinivas Somarowthu⁴, Anna Marie Pyle^{5,6,7}, Marco De Vivo^{1,*}, Marco Marcia^{2,*}

*To whom correspondence should be addressed. E-mail: marco.devivo@iit.it; mmarcia@embl.fr

We would like to thank the editor and reviewer #4 for their further comments on our manuscript. Here, we report our response to reviewer #4. Our detailed point-by-point response to the editor is reported in the cover letter, as requested.

REVIEWER #4:

The authors have addressed all my additional comments on the manuscript, and I am satisfied with their revisions.

Thank you.